# FASTER BINARY EMBEDDINGS FOR PRESERVING EUCLIDEAN DISTANCES

**Jinjie Zhang & Rayan Saab** [*]
Department of Mathematics, Halıcıoğlu Data Science Institute
University of California San Diego
{jiz003, rsaab}@ucsd.edu

## ABSTRACT

We propose a fast, distance-preserving, binary embedding algorithm to transform a high-dimensional dataset $\mathcal{T} \subseteq \mathbb{R}^n$ into binary sequences in the cube $\{\pm 1\}^m$. When $\mathcal{T}$ consists of well-spread (i.e., non-sparse) vectors, our embedding method applies a stable noise-shaping quantization scheme to $\mathbf{A}\boldsymbol{x}$ where $\mathbf{A} \in \mathbb{R}^{m \times n}$ is a sparse Gaussian random matrix. This contrasts with most binary embedding methods, which usually use $\boldsymbol{x} \mapsto \mathrm{sign}(\mathbf{A}\boldsymbol{x})$ for the embedding. Moreover, we show that Euclidean distances among the elements of $\mathcal{T}$ are approximated by the $\ell_1$ norm on the images of $\{\pm 1\}^m$ under a fast linear transformation. This again contrasts with standard methods, where the Hamming distance is used instead. Our method is both fast and memory efficient, with time complexity $O(m)$ and space complexity $O(m)$ on well-spread data. When the data is not well-spread, we show that the approach still works provided that data is transformed via a Walsh-Hadamard matrix, but now the cost is $O(n \log n)$ per data point. Further, we prove that the method is accurate and its associated error is comparable to that of a continuous valued Johnson-Lindenstrauss embedding plus a quantization error that admits a polynomial decay as the embedding dimension $m$ increases. Thus the length of the binary codes required to achieve a desired accuracy is quite small, and we show it can even be compressed further without compromising the accuracy. To illustrate our results, we test the proposed method on natural images and show that it achieves strong performance.

## 1 INTRODUCTION

Analyzing large data sets of high-dimensional raw data is usually computationally demanding and memory intensive. As a result, it is often necessary as a preprocessing step to transform data into a lower-dimensional space while approximately preserving important geometric properties, such as pairwise $\ell_2$ distances. As a critical result in dimensionality reduction, the Johnson-Lindenstrauss (JL) lemma (Johnson & Lindenstrauss, 1984) guarantees that every finite set $\mathcal{T} \subseteq \mathbb{R}^n$ can be (linearly) mapped to a $m = O(\epsilon^{-2} \log(|\mathcal{T}|))$ dimensional space in such a way that all pairwise distances are preserved up to an $\epsilon$-Lipschitz distortion. Additionally, there are many significant results to speed up the JL transform by introducing fast embeddings, e.g. (Ailon & Chazelle, 2009; Ailon & Liberty, 2013; Krahmer & Ward, 2011; Nelson et al., 2014), or by using sparse matrices (Kane & Nelson, 2014; 2010; Clarkson & Woodruff, 2017). Such fast embeddings can usually be computed in $O(n \log n)$ versus the $O(mn)$ time complexity of JL transforms that rely on unstructured dense matrices.

### 1.1 RELATED WORK

To further reduce memory requirements, progress has been made in *nonlinearly* embedding high-dimensional sets $\mathcal{T} \subseteq \mathbb{R}^n$ to the binary cube $\{-1, 1\}^m$ with $m \ll n$, a process known as binary embedding. Provided that $d_1(\cdot, \cdot)$ is a metric on $\mathbb{R}^n$, a distace preserving binary embedding is a map

---

[*]The Python source code of our paper: https://github.com/jayzhang0727/Faster-Binary-Embeddings-for-Preserving-Euclidean-Distances.git

$f : \mathcal{T} \to \{-1, 1\}^m$ and a function $d_2(\cdot, \cdot)$ on $\{-1, 1\}^m \times \{-1, 1\}^m$ to approximate distances, i.e.,

$$|d_2(f(\boldsymbol{x}), f(\boldsymbol{y})) - d_1(\boldsymbol{x}, \boldsymbol{y})| \le \alpha, \quad \text{for } \forall \boldsymbol{x}, \boldsymbol{y} \in \mathcal{T}. \tag{1}$$

The potential dimensionality reduction ($m \ll n$) and 1-bit representation per dimension imply that storage space can be considerably reduced and downstream applications like learning and retrieval can happen directly using bitwise operations. Most existing nonlinear mappings $f$ in (1) are generated using simple memory-less scalar quantization (MSQ). For example, given a set of unit vectors $\mathcal{T} \subseteq \mathbb{S}^{n-1}$ with finite size $|\mathcal{T}|$, consider the map

$$\boldsymbol{q_x} := f(\boldsymbol{x}) = \text{sign}(\mathbf{G}\boldsymbol{x}) \tag{2}$$

where $\mathbf{G} \in \mathbb{R}^{m \times n}$ is a standard Gaussian random matrix and $\text{sign}(\cdot)$ returns the element-wise sign of its argument. Let $d_1(\boldsymbol{x}, \boldsymbol{y}) = \frac{1}{\pi} \arccos(\|\boldsymbol{x}\|_2^{-1} \|\boldsymbol{y}\|_2^{-1} \langle \boldsymbol{x}, \boldsymbol{y} \rangle)$ be the normalized angular distance and $d_2(\boldsymbol{q_x}, \boldsymbol{q_y}) = \frac{1}{2m} \|\boldsymbol{q_x} - \boldsymbol{q_y}\|_1$ be the normalized Hamming distance. Then, Yi et al. (2015) show that (1) holds with probability at least $1 - \eta$ if $m \gtrsim \alpha^{-2} \log(|\mathcal{T}|/\eta)$, so one can approximate geodesic distances with normalized Hamming distances. While this approach achieves optimal bit complexity (up to constants) (Yi et al., 2015), it has been observed in practice that $m$ is usually around $O(n)$ to guarantee reasonable accuracy (Gong et al., 2013; Sánchez & Perronnin, 2011; Yu et al., 2014). Much like linear JL embedding techniques admit fast counterparts, fast binary embedding algorithms have been developed to significantly reduce the runtime of binary embeddings (Gong et al., 2012b; Liu et al., 2011; Gong et al., 2012a; 2013; Li et al., 2011; Raginsky & Lazebnik, 2009). Indeed, fast JL transforms (FJLT) and Gaussian Toeplitz matrices (Yi et al., 2015), structured hashed projections (Choromanska et al., 2016), iterative quantization (Gong et al., 2012b), bilinear projection (Gong et al., 2013), circulant binary embedding (Yu et al., 2014; Dirksen & Stollenwerk, 2018; 2017; Oymak et al., 2017; Kim et al., 2018), sparse projection (Xia et al., 2015), and fast orthogonal projection (Zhang et al., 2015) have all been considered.

These methods can decrease time complexity to $O(n \log n)$ operations per embedding, but still suffer from some important drawbacks. Notably, due to the sign function, these algorithms completely discard all magnitude information, as $\text{sign}(\mathbf{A}\boldsymbol{x}) = \text{sign}(\mathbf{A}(\alpha \boldsymbol{x}))$ for all $\alpha > 0$. So, all points in the same direction embed to the same binary vector and cannot be distinguished. Even if one settles for recovering geodesic distances, using the sign function in (2) is an instance of MSQ so the estimation error $\alpha$ in (1) decays slowly as the number of bits $m$ increases (Yi et al., 2015).

In addition to the above data independent approaches, there are data dependent embedding methods for distance recovery, including product quantization (Jegou et al., 2010; Ge et al., 2013), LSH-based methods (Andoni & Indyk, 2006; Shrivastava & Li, 2014; Datar et al., 2004) and iterative quantization (Gong et al., 2012c). Their accuracy, which can be excellent, nevertheless depends on the underlying distribution of the input dataset. Moreover, they may be associated with larger time and space complexity for embedding the data. For example, product quantization performs $k$-means clustering in each subspace to find potential centroids and stores associated lookup tables. LSH-based methods need random shifts and dense random projections to quantize each input data point.

Recently Huynh & Saab (2020) resolved these issues by replacing the simple sign function with a Sigma-Delta ($\Sigma\Delta$) quantization scheme, or alternatively other noise-shaping schemes (see (Chou & Güntürk, 2016)) whose properties will be discussed in Section 3. They use the binary embedding

$$\boldsymbol{q_x} := Q(\mathbf{DB}\boldsymbol{x}) \tag{3}$$

where $Q$ is now a stable $\Sigma\Delta$ quantization scheme, $\mathbf{D} \in \mathbb{R}^{m \times m}$ is a diagonal matrix with random signs, and $\mathbf{B} \in \mathbb{R}^{m \times n}$ are specific structured random matrices. To give an example of $\Sigma\Delta$ quantization in this context, consider $\boldsymbol{w} := \mathbf{DB}\boldsymbol{x}$. Then the simplest $\Sigma\Delta$ scheme computes $\boldsymbol{q_x}$ via the following iteration, run for $i = 1, ..., m$:

$$\begin{cases} u_0 = 0, \\ \boldsymbol{q_x}(i) = \text{sign}(w_i + u_{i-1}), \\ u_i = u_{i-1} + w_i - q_i. \end{cases} \tag{4}$$

The choices of $\mathbf{B}$ in (Huynh & Saab, 2020) allow matrix vector multiplication to be implemented using the fast Fourier transform. Then the original Euclidean distance $\|\boldsymbol{x} - \boldsymbol{y}\|_2$ can be recovered via a pseudo-metric on the quantized vectors given by

$$d_{\widetilde{\boldsymbol{V}}}(\boldsymbol{q_x}, \boldsymbol{q_y}) := \|\widetilde{\boldsymbol{V}}(\boldsymbol{q_x} - \boldsymbol{q_y})\|_2 \tag{5}$$

where $\widetilde{V} \in \mathbb{R}^{p \times m}$ is a "normalized condensation operator", a sparse matrix that can be applied fast (see Section 3). Regarding the complexity of applying (3) to a single $x \in \mathbb{R}^n$, note that $x \mapsto \mathbf{DB}x$ has time complexity $O(n \log n)$ while the quantization map needs $O(m)$ time and results in an $m$ bit representation. So when $m \leq n$, the total time complexity for (3) is around $O(n \log n)$.

## 1.2 METHODS AND CONTRIBUTIONS

We extend these results by replacing $\mathbf{DB}$ in (3) by a sparse Gaussian matrix $\mathbf{A} \in \mathbb{R}^{m \times n}$ so that now

$$q_x := Q(\mathbf{A}x). \tag{6}$$

Given scaled high-dimensional data $\mathcal{T} \subset \mathbb{R}^n$ contained in the $\ell_2$ ball $B_2^n(\kappa)$ with radius $\kappa$, we put forward Algorithm 1 to generate binary sequences and Algorithm 2 to compute estimates of the Euclidean distances between elements of $\mathcal{T}$ via an $\ell_1$-norm rather than $\ell_2$-norm. The contribution of this work is threefold. First, we prove Theorem 1.1 quantifying the performance of our algorithms.

---

**Algorithm 1:** Fast Binary Embedding for Finite $\mathcal{T}$

**Input:** $\mathcal{T} = \{x^{(j)}\}_{j=1}^k \subseteq B_2^n(\kappa)$       ▷ Data points in $\ell_2$ ball
Generate $\mathbf{A} \in \mathbb{R}^{m \times n}$ as in Definition 2.2     ▷ Sparse Gaussian matrix $\mathbf{A}$
**for** $j \leftarrow 1$ **to** $k$ **do**
  $z^{(j)} \leftarrow \mathbf{A}x^{(j)}$
  $q^{(j)} = Q(z^{(j)})$     ▷ Stable $\Sigma\Delta$ quantizer $Q$ as in (4), or more generally (21).
**Output:** Binary sequences $\mathcal{B} = \{q^{(j)}\}_{j=1}^k \subseteq \{-1, 1\}^m$

---

**Algorithm 2:** $\ell_2$ Norm Distance Recovery

**Input:** $q^{(i)}, q^{(j)} \in \mathcal{B}$      ▷ Binary sequences produced by Algorithm 1
$y^{(i)} \leftarrow \widetilde{V}q^{(i)}$         ▷ Condense the components of $q$
$y^{(j)} \leftarrow \widetilde{V}q^{(j)}$
**Output:** $\|y^{(i)} - y^{(j)}\|_1$      ▷ Approximation of $\|x^{(i)} - x^{(j)}\|_2$

---

**Theorem 1.1** (Main result). *Let $\mathcal{T} \subseteq \mathbb{R}^n$ be a finite, appropriately scaled set with elements satisfying $\|x\|_\infty = O(n^{-1/2}\|x\|_2)$ and $\|x\|_2 \leq \kappa < 1$. If $m \gtrsim p := \Omega(\epsilon^{-2}\log(|\mathcal{T}|^2/\delta))$ and $r \geq 1$ is the integer order of $Q$, then with probability $1 - 2\delta$ on the draw of the sparse Gaussian matrix $\mathbf{A}$, the following holds uniformly over all $x, y$ in $\mathcal{T}$: Embedding $x, y$ into $\{-1, 1\}^m$ using Algorithm 1, and estimating the associated distance between them using Algorithm 2 yields the error bound*

$$\left| d_{\widetilde{V}}(q_x, q_y) - \|x - y\|_2 \right| \leq c \left( \frac{m}{p} \right)^{-r+1/2} + \epsilon \|x - y\|_2$$

*where $c > 0$ is a constant.*

Theorem 1.1 yields an approximation error bounded by two components, one due to quantization and another that resembles the error from a *linear* JL embedding into a $p$-dimensional space. The latter part is essentially proportional to $p^{-1/2}$, while the quantization component decays polynomially fast in $m$, and can be made harmless by increasing $m$. Moreover, the number of bits $m \gtrsim \epsilon^{-2}\log(|\mathcal{T}|)$ achieves the optimal bit complexity required by any oblivious random embedding that preserves Euclidean or squared Euclidean distance, see Theorem 4.1 in (Dirksen & Stollenwerk, 2020). Theorem 4.2 is a more precise version of Theorem 1.1, with all quantizers, and scaling parameters specified explicitly, and with a potential modification to $\mathbf{A}$ that enables the result to hold for arbitrary (not necessarily well-spread) finite $\mathcal{T}$, at the cost of increasing the computational complexity of embedding a point to $O(n \log n)$. We also note that if the data did not satisfy the scaling assumption of Theorems 1.1 and 4.2, then one can replace $\{-1, 1\}$ by $\{-C, C\}$, and the quantization error would scale by $C$.

Second, due to the sparsity of $\mathbf{A}$, (6) can be computed much faster than (3), when restricting our results to "well-spread" vectors $x$, i.e., those that are not sparse. On the other hand, in Section 5, we show that Algorithm 1 achieves $O(m)$ time and space complexity in contrast with the common $O(n \log n)$ runtime of fast binary embeddings, e.g., (Gong et al., 2013; Yi et al., 2015; Yu et al.,

2014; Dirksen & Stollenwerk, 2018; 2017; Huynh & Saab, 2020) that rely on fast JL transforms or circulant matrices. Meanwhile, Algorithm 2 requires only $O(m)$ runtime.

Third, Definition 2.3 shows that $\widetilde{V}$ is sparse and essentially populated by integers bounded by $(m/p)^r$ where $r, m, p$ are as in Theorem 1.1. In Section 5, we note that each $\boldsymbol{y}^{(i)} = \widetilde{\boldsymbol{V}} \boldsymbol{q}^{(i)}$ (and the distance query), can be represented by $O(p \log_2(m/p))$ bits, instead of $m$ bits, without affecting the reconstruction accuracy. This is a consequence of using the $\ell_1$-norm in Algorithm 2. Had we instead used an $\ell_2$-norm, we would have required $O(p(\log_2(m/p))^2)$ bits.

Finally, we remark that while the assumption that the vectors $x$ are well-spread (i.e. $\|\boldsymbol{x}\|_\infty = O(n^{-1/2}\|\boldsymbol{x}\|_2)$) may appear restrictive, there are important instances where it holds. Natural images seem to be one such case, as are random Fourier features (Rahimi & Recht, 2007). Similarly, Gaussian (and other subgaussian) random vectors satisfy a slightly weakened $\|\boldsymbol{x}\|_\infty = O(\log(n)n^{-1/2}\|\boldsymbol{x}\|_2)$ assumption with high probability, and one can modify our construction by slightly reducing the sparsity of $\mathbf{A}$ (and slightly increasing the computational cost) to handle such vectors. On the other hand, if the data simply does not satisfy such an assumption, one can still apply Theorem 4.2 part (ii), but now the complexity of embedding a point is $O(n \log n)$.

## 2 PRELIMINARIES

### 2.1 NOTATION AND DEFINITIONS

Throughout, $f(n) = O(g(n))$ and $f(n) = \Omega(g(n))$ mean that $|f(n)|$ is bounded above and below respectively by a positive function $g(n)$ up to constants asymptotically; that is, $\limsup_{n \to \infty} \frac{|f(n)|}{g(n)} < \infty$. Similarly, we use $f(n) = \Theta(g(n))$ to denote that $f(n)$ is bounded both above and below by a positive function $g(n)$ up to constants asymptotically. We next define operator norms.

**Definition 2.1.** Let $\alpha, \beta \in [1, \infty]$ be integers. The $(\alpha, \beta)$ operator norm of $\boldsymbol{K} \in \mathbb{R}^{m \times n}$ is $\|\boldsymbol{K}\|_{\alpha,\beta} = \max_{x \neq 0} \frac{\|\boldsymbol{K}\boldsymbol{x}\|_\beta}{\|\boldsymbol{x}\|_\alpha}$.

We now introduce some notation and definitions that are relevant to our construction.

**Definition 2.2** (Sparse Gaussian random matrix). Let $\mathbf{A} = (\mathrm{a}_{ij}) \in \mathbb{R}^{m \times n}$ be a random matrix with i.i.d. entries such that $\mathrm{a}_{ij}$ is 0 with probability $1 - s$ and is drawn from $\mathcal{N}(0, \frac{1}{s})$ with probability $s$.

We adopt the definition of a condensation operator of Chou & Güntürk (2016); Huynh & Saab (2020).

**Definition 2.3** (Condensation operator). Let $p, r, \lambda$ be fixed positive integers such that $\lambda = r\widetilde{\lambda} - r + 1$ for some integer $\widetilde{\lambda}$. Let $m = \lambda p$ and $\boldsymbol{v}$ be a row vector in $\mathbb{R}^\lambda$ whose entry $v_j$ is the $j$-th coefficient of the polynomial $(1 + z + \ldots + z^{\widetilde{\lambda}-1})^r$. Define the condensation operator $\boldsymbol{V} \in \mathbb{R}^{p \times m}$ by

$$\boldsymbol{V} = \boldsymbol{I}_p \otimes \boldsymbol{v} = \begin{bmatrix} \boldsymbol{v} & & \\ & \ddots & \\ & & \boldsymbol{v} \end{bmatrix}.$$

For example, when $r = 1$, $\lambda = \widetilde{\lambda}$, and $\boldsymbol{v} \in \mathbb{R}^\lambda$ is simply the vector of all ones. The normalized condensation operator is given by

$$\widetilde{\boldsymbol{V}} = \frac{\sqrt{\pi/2}}{p\|\boldsymbol{v}\|_2} \boldsymbol{V}.$$

The fast JL transform was first studied by Ailon & Chazelle (2009). It admits many variants and improvements, e.g. (Krahmer & Ward, 2011; Matoušek, 2008). The idea is that given any $\boldsymbol{x} \in \mathbb{R}^n$ we use a fast "Fourier-like" transform, like the Walsh-Hadamard transform, to distribute the total mass (i.e. $\|\boldsymbol{x}\|_2$) of $\boldsymbol{x}$ relatively evenly to its coordinates.

**Definition 2.4** (FJLT). The fast JL transform can be obtained by

$$\boldsymbol{\Phi} := \mathbf{A}\boldsymbol{H}\boldsymbol{D} \in \mathbb{R}^{m \times n}. \tag{7}$$

Here, $\mathbf{A} \in \mathbb{R}^{m \times n}$ is a sparse Gaussian random matrix, as in Definition 2.2, while $\boldsymbol{H} \in \mathbb{R}^{n \times n}$ is a normalized Walsh-Hadamard matrix defined by $H_{ij} = n^{-1/2}(-1)^{\langle i-1, j-1 \rangle}$ where $\langle i, j \rangle$ is the bitwise dot product of the binary representations of the numbers $i$ and $j$. Finally, $\mathbf{D} \in \mathbb{R}^{n \times n}$ is diagonal with diagonal entries drawn independently from $\{-1, 1\}$ with probability $1/2$ for each.

## 2.2 CONDENSED JOHNSON-LINDENSTRAUSS TRANSFORMS

**Definition 2.5.** When $\widetilde{V}$ is a condensation operator, and $\mathbf{A}$ is a sparse Gaussian, we refer to $\widetilde{V}\mathbf{A}$ as a condensed sparse JL transform (CSJLT). When $\mathbf{A}$ is replaced by $\mathbf{\Phi}$ as in Definition 2.4 we refer to $\widetilde{V}\mathbf{\Phi}$ as a condensed fast JL transform (CFJLT).

The definition above is justified by the following lemma (see Appendix B for the proof).

**Lemma 2.6** (CJLT lemma). *Let $\mathcal{T}$ be a finite subset of $\mathbb{R}^n$, $\lambda \in \mathbb{N}$, $\epsilon \in (0, \frac{1}{2})$, $\delta \in (0, 1)$, $p = O(\epsilon^{-2} \log(|\mathcal{T}|^2/\delta)) \in \mathbb{N}$ and $m = \lambda p$. Let $\widetilde{V} \in \mathbb{R}^{p \times m}$ be as in Definition 2.3, $\mathbf{A} \in \mathbb{R}^{m \times n}$ be the sparse Gaussian matrix in Definition 2.2 with $s = \Theta(\epsilon^{-1} n^{-1}(\|\boldsymbol{v}\|_\infty/\|\boldsymbol{v}\|_2)^2) \leq 1$, and $\mathbf{\Phi} = \mathbf{A}\boldsymbol{H}\boldsymbol{D} \in \mathbb{R}^{m \times n}$ be the FJLT in Definition 2.4 with $s = \Theta(\epsilon^{-1} n^{-1}(\|\boldsymbol{v}\|_\infty/\|\boldsymbol{v}\|_2)^2 \log n) \leq 1$. If $\mathcal{T}$ consists of well-spread vectors, that is, $\|\boldsymbol{x}\|_\infty = O(n^{-1/2}\|\boldsymbol{x}\|_2)$ for all $\boldsymbol{x} \in \mathcal{T}$, then*

$$\left| \|\widetilde{V}\mathbf{A}(\boldsymbol{x} - \boldsymbol{y})\|_1 - \|\boldsymbol{x} - \boldsymbol{y}\|_2 \right| \leq \epsilon \|\boldsymbol{x} - \boldsymbol{y}\|_2 \tag{8}$$

*holds uniformly for all $\boldsymbol{x}, \boldsymbol{y} \in \mathcal{T}$ with probability at least $1 - \delta$. If $\mathcal{T}$ is finite but arbitrary, then*

$$\left| \|\widetilde{V}\mathbf{\Phi}(\boldsymbol{x} - \boldsymbol{y})\|_1 - \|\boldsymbol{x} - \boldsymbol{y}\|_2 \right| \leq \epsilon \|\boldsymbol{x} - \boldsymbol{y}\|_2 \tag{9}$$

*holds uniformly for all $\boldsymbol{x}, \boldsymbol{y} \in \mathcal{T}$ with probability at least $1 - \delta$.*

So $\mathcal{T} \subseteq \mathbb{R}^n$ is embedded into $\mathbb{R}^p$ with pairwise distances distorted at most $\epsilon$, where $p = O(\epsilon^{-2} \log |\mathcal{T}|)$ as one would expect from a JL embedding. This will be needed to guarantee the accuracy associated with our embeddings algorithms. Note that the bound on $p$ does not require extra logarithmic factors, in contrast to the bound $O(\epsilon^{-2} \log |\mathcal{T}| \log^4 n)$ in (Huynh & Saab, 2020).

## 3 SIGMA-DELTA QUANTIZATION

An $r$-th order $\Sigma\Delta$ quantizer $Q^{(r)} : \mathbb{R}^m \to \mathcal{A}^m$ maps an input signal $\boldsymbol{y} = (y_i)_{i=1}^m \in \mathbb{R}^m$ to a quantized sequence $\boldsymbol{q} = (q_i)_{i=1}^m \in \mathcal{A}^m$ via a quantization rule $\rho$ and the following iterations

$$\begin{cases} u_0 = u_{-1} = \ldots = u_{1-r} = 0, \\ q_i = Q(\rho(y_i, u_{i-1}, \ldots, u_{i-r})) \quad \text{for } i = 1, 2, \ldots, m, \\ \boldsymbol{P}^r \boldsymbol{u} = \boldsymbol{y} - \boldsymbol{q} \end{cases} \tag{10}$$

where $Q(y) = \arg\min_{v \in \mathcal{A}} |y - v|$ is the scalar quantizer related to alphabet $\mathcal{A}$ and $\boldsymbol{P} \in \mathbb{R}^{m \times m}$ is the first order difference matrix defined by

$$P_{ij} = \begin{cases} 1 & \text{if } i = j, \\ -1 & \text{if } i = j + 1, \\ 0 & \text{otherwise.} \end{cases}$$

Note that (10) is amenable to an iterative update of the state variables $u_i$ as

$$\boldsymbol{P}^r \boldsymbol{u} = \boldsymbol{y} - \boldsymbol{q} \iff u_i = \sum_{j=1}^r (-1)^{j-1} \binom{r}{j} u_{i-j} + y_i - q_i, \quad i = 1, 2, \ldots, m. \tag{11}$$

**Definition 3.1.** A quantization scheme is *stable* if there exists $\mu > 0$ such that for each input with $\|\boldsymbol{y}\|_\infty \leq \mu$, the state vector $\boldsymbol{u} \in \mathbb{R}^m$ satisfies $\|\boldsymbol{u}\|_\infty \leq C$. Crucially, $\mu$ and $C$ do not depend on $m$.

Stability heavily depends on the choice of quantization rule and is difficult to guarantee for arbitrary $\rho$ in (10) when the alphabet is small, as is the case of 1-bit quantization where $\mathcal{A} = \{\pm 1\}$. When $r = 1$ and $\mathcal{A} = \{\pm 1\}$, the simplest stable $\Sigma\Delta$ scheme $Q^{(1)} : \mathbb{R}^m \to \mathcal{A}^m$ is equipped with the greedy quantization rule $\rho(y_i, u_{i-1}) := u_{i-1} + y_i$ giving the simple iteration (4) from the introduction, albeit with $y_i$ replacing $w_i$. A description of the design and properties of stable $Q^{(r)}$ with $r \geq 2$ can be found in Appendix C.

## 4  MAIN RESULTS

The ingredients that make our construction work are a JL embedding followed by $\Sigma\Delta$ quantization. Together these embed points into $\{\pm1\}^m$, but it remains to define a pseudometric so that we may approximate Euclidean distances by distances on the cube. We now define this pseudometric.

**Definition 4.1.** Let $\mathcal{A}^m = \{\pm1\}^m$ and let $\boldsymbol{V} \in \mathbb{R}^{p\times m}$ with $p \le m$. We define $d_{\boldsymbol{V}}$ on $\mathcal{A}^m \times \mathcal{A}^m$ as

$$d_{\boldsymbol{V}}(\boldsymbol{q}_1, \boldsymbol{q}_2) = \|\boldsymbol{V}(\boldsymbol{q}_1 - \boldsymbol{q}_2)\|_1 \quad \forall\, \boldsymbol{q}_1, \boldsymbol{q}_2 \in \mathcal{A}^m.$$

We now present our main result, a more technical version of Theorem 1.1, proved in Appendix D.

**Theorem 4.2** (Main result). *Let* $\lambda,\ r \in \mathbb{N}$, $\epsilon \in (0, \frac{1}{2})$, $\delta \in (0,1)$, $\beta = \Omega(\log(|\mathcal{T}|/\delta)) > 0$, $\mu \in (0,1)$, $p = \Omega(\epsilon^{-2}\log(|\mathcal{T}|^2/\delta)) \in \mathbb{N}$, *and* $m = \lambda p$. *Let* $\widetilde{\boldsymbol{V}} \in \mathbb{R}^{p\times m}$ *be as in Definition 2.3,* $\boldsymbol{A} \in \mathbb{R}^{m\times n}$ *be the sparse Gaussian matrix in Definition 2.2 with* $s = \Theta(\epsilon^{-1}n^{-1}(\|\boldsymbol{v}\|_\infty/\|\boldsymbol{v}\|_2)^2) \le 1$, *and* $\boldsymbol{\Phi}$ *be the FJLT in Definition 2.4 with* $s = \Theta(\epsilon^{-1}n^{-1}(\|\boldsymbol{v}\|_\infty/\|\boldsymbol{v}\|_2)^2 \log n) \le 1$.

*Let* $\mathcal{T}$ *be a finite subset of* $B_2^n(\kappa) := \{\boldsymbol{x} \in \mathbb{R}^n : \|\boldsymbol{x}\|_2 \le \kappa\}$ *and suppose that*

$$\kappa \le \frac{\mu}{2\sqrt{\beta + \log(2m)}}.$$

*Defining the embedding maps* $f_1 : \mathcal{T} \to \{\pm1\}^m$ *by* $f_1 = Q^{(r)} \circ \boldsymbol{A}$ *and* $f_2 : \mathcal{T} \to \{\pm1\}^m$ *by* $f_2 = Q^{(r)} \circ \boldsymbol{\Phi}$, *there exists a constant* $C(\mu, r)$ *such that the following are true:*

*(i) If the elements of* $\mathcal{T}$ *satisfy* $\|\boldsymbol{x}\|_\infty = O(n^{-1/2}\|\boldsymbol{x}\|_2)$, *then the bound*

$$\left| d_{\widetilde{\boldsymbol{V}}}(f_1(\boldsymbol{x}), f_1(\boldsymbol{y})) - \|\boldsymbol{x} - \boldsymbol{y}\|_2 \right| \le C(\mu, r)\lambda^{-r+1/2} + \epsilon\|\boldsymbol{x} - \boldsymbol{y}\|_2 \tag{12}$$

*holds uniformly for all* $\boldsymbol{x}, \boldsymbol{y} \in \mathcal{T}$ *with probability exceeding* $1 - \delta - |\mathcal{T}|e^{-\beta}$.

*(ii) On the other hand, for arbitrary* $\mathcal{T} \subset B_2^n(\kappa)$

$$\left| d_{\widetilde{\boldsymbol{V}}}(f_2(\boldsymbol{x}), f_2(\boldsymbol{y})) - \|\boldsymbol{x} - \boldsymbol{y}\|_2 \right| \le C(\mu, r)\lambda^{-r+1/2} + \epsilon\|\boldsymbol{x} - \boldsymbol{y}\|_2 \tag{13}$$

*holds uniformly for any* $\boldsymbol{x}, \boldsymbol{y} \in \mathcal{T}$ *with probability exceeding* $1 - \delta - 2|\mathcal{T}|e^{-\beta}$.

Under the assumptions of Theorem 4.2, we have

$$\epsilon = O\left(\sqrt{\frac{\log(|\mathcal{T}|^2/\delta)}{p}}\right) \lesssim \frac{1}{\sqrt{p}}. \tag{14}$$

By (12), (13) and (14), we have that with high probability the inequality

$$\begin{aligned}
\left| d_{\widetilde{\boldsymbol{V}}}(f_i(\boldsymbol{x}), f_i(\boldsymbol{y})) - \|\boldsymbol{x} - \boldsymbol{y}\|_2 \right| &\le C(\mu, r)\left(\frac{m}{p}\right)^{-r+1/2} + \epsilon\|\boldsymbol{x} - \boldsymbol{y}\|_2 \\
&\le C(\mu, r)\left(\frac{m}{p}\right)^{-r+1/2} + 2\kappa\epsilon \\
&\le C(\mu, r)\left(\frac{m}{p}\right)^{-r+1/2} + \frac{\mu}{\sqrt{\beta + \log(2m)}} \cdot \frac{C_2}{\sqrt{p}} \tag{15}
\end{aligned}$$

holds uniformly for $\boldsymbol{x}, \boldsymbol{y} \in \mathcal{T}$. The first error term in (15) results from $\Sigma\Delta$ quantization while the second error term is caused by the CJLT. So the term $O((m/p)^{-r+1/2})$ dominates when $\lambda = m/p$ is small. If $m/p$ is sufficiently large, the second term $O(1/\sqrt{p})$ becomes dominant.

## 5  COMPUTATIONAL AND SPACE COMPLEXITY

In this section, we assume that $\mathcal{T} = \{\boldsymbol{x}^{(j)}\}_{j=1}^k \subseteq \mathbb{R}^n$ consists of well-spread vectors. Moreover, we will focus on stable $r$-th order $\Sigma\Delta$ schemes $Q^{(r)} : \mathbb{R}^m \to \mathcal{A}^m$ with $\mathcal{A} = \{-1, 1\}$. By Definition 2.3, when $r = 1$ we have $\boldsymbol{v} = (1, 1, \ldots, 1) \in \mathbb{R}^\lambda$, while when $r = 2$, $\boldsymbol{v} = (1, 2, \ldots, \widetilde{\lambda} - 1, \widetilde{\lambda}, \widetilde{\lambda} -$

$1, \ldots, 2, 1) \in \mathbb{R}^\lambda$. In general, $\|\boldsymbol{v}\|_\infty / \|\boldsymbol{v}\|_2 = O(\lambda^{-1/2})$ holds for all $r \in \mathbb{N}$. We also assume that $s = \Theta(\epsilon^{-1} n^{-1} (\|\boldsymbol{v}\|_\infty / \|\boldsymbol{v}\|_2)^2) = \Theta(\epsilon^{-1} n^{-1} \lambda^{-1}) \le 1$ as in Theorem 4.2. We consider $b$-bit floating-point or fixed-point representations for numbers. Both entail the same computational complexity for computing sums and products of two numbers. Addition and subtraction require $O(b)$ operations while multiplication and division require $\mathcal{M}(b) = O(b^2)$ operations via "standard" long multiplication and division. Multiplication and division can be done more efficiently, particularly for large integers and the best known methods (and best possible up to constants) have complexity $\mathcal{M}(b) = O(b \log b)$ (Harvey & Van Der Hoeven, 2019). We also assume random access to the coordinates of our data points.

**Embedding complexity.** For each data point $\boldsymbol{x}^{(j)} \in \mathcal{T}$, one can use Algorithm 1 to quantize it. Since $\mathbf{A}$ has sparsity constant $s = \Theta(\epsilon^{-1} n^{-1} \lambda^{-1})$ and $\epsilon^{-1} = O(p^{1/2})$ by (14), and since $\lambda = m/p$, computing $\mathbf{A}\boldsymbol{x}^{(j)}$ needs $O(snm) = O(\lambda^{-1} \epsilon^{-1} m) = O(p^{3/2})$ time. Additionally, it takes $O(m)$ time to quantize $\mathbf{A}\boldsymbol{x}^{(j)}$ based on (21). When $p^{3/2} \le m$, Algorithm 1 can be executed in $O(m)$ for each $\boldsymbol{x}^{(j)}$. Because $\mathbf{A}$ has $O(snm) = O(m)$ nonzero entries, the space complexity is $O(m)$ bits per data point. Note that the big $O$ notation here hides the space complexity dependence on the bit-depth $b$ of the fixed or floating point representation of the entries of $\mathbf{A}$ and $\boldsymbol{x}^{(j)}$. This clearly has no effect on the storage space needed for each $\boldsymbol{q}^{(j)}$, which is exactly $m$ bits.

**Complexity of distance estimation.** If one does not use embedding methods, storing $\mathcal{T}$ directly, i.e., by representing the coefficients of each $\boldsymbol{x}^{(j)}$ by $b$ bits requires $knb$ bits. Moreover, the resulting computational complexity of estimating $\|\boldsymbol{x} - \boldsymbol{y}\|_2^2$ where $\boldsymbol{x}, \boldsymbol{y} \in \mathcal{T}$ is $O(n\mathcal{M}(b))$. On the other hand, suppose we obtain binary sequences $\mathcal{B} = \{\boldsymbol{q}^{(j)}\}_{j=1}^k \subseteq \mathcal{A}^m$ by performing Algorithm 1 on $\mathcal{T}$. Using our method with accuracy guaranteed by Theorem 4.2, high-dimensional data points $\mathcal{T} \subseteq \mathbb{R}^n$ are now transformed into short binary sequences, which only require $km$ bits of storage instead of $knb$ bits. Algorithm 2 can be applied to recover the pairwise $\ell_2$ distances. Note that $\widetilde{V}$ is the normalization of an integer valued matrix $\boldsymbol{V} = \boldsymbol{I}_p \otimes \boldsymbol{v}$ (by Definition 2.3) and $\boldsymbol{q}^{(i)} \in \mathcal{A}^m$ is a binary vector. So, by storing the normalization factor separately, we can ignore it when considering runtime and space complexity. Thus we observe:

1. The number of bits needed to represent each entry of $\boldsymbol{v}$ is at most $\log_2(\|\boldsymbol{v}\|_\infty) \approx (r - 1) \log_2 \lambda = O(\log_2 \lambda)$ when $r > 1$ and $O(1)$ when $r = 1$. So the computation of $\boldsymbol{y}^{(i)} = \widetilde{V} \boldsymbol{q}^{(i)} \in \mathbb{R}^p$ only involves $m$ additions or subtractions of integers represented by $O(\log_2 \lambda)$ bits and thus the time complexity in computing $\boldsymbol{y}^{(i)}$ is $O(m \log_2 \lambda)$.

2. Each of the $p$ entries of $\boldsymbol{y}^{(i)}$ is the sum of $\lambda$ terms each bounded by $\lambda^{r-1}$. We can store $\boldsymbol{y}^{(i)}$ in $O(p \log_2 \lambda)$ bits.

3. Computing $\|\boldsymbol{y}^{(i)} - \boldsymbol{y}^{(j)}\|_1$ needs $O(p \log_2 \lambda)$ time and $O(p \log_2 \lambda)$ bits.

So we use $O(p \log_2 \lambda)$ bits to recover each pairwise distance $\|\boldsymbol{x}^{(i)} - \boldsymbol{x}^{(j)}\|_2$ in $O(m \log_2 \lambda)$ time.

| Method | Time | Space | Storage | Query Time |
|---|---|---|---|---|
| Gaussian Toeplitz (Yi et al., 2015) | $O(n \log n)$ | $O(n)$ | $O(m)$ | $O(m)$ |
| Bilinear (Gong et al., 2013) | $O(n\sqrt{m})$ | $O(\sqrt{mn})$ | $O(m)$ | $O(m)$ |
| Circulant (Yu et al., 2014) | $O(n \log n)$ | $O(n)$ | $O(m)$ | $O(m)$ |
| BOE or PCE$^\star$ (Huynh & Saab, 2020) | $O(n \log n)$ | $O(n)$ | $O(p \log_2 \lambda)$ | $O(p\mathcal{M}(\log_2 \lambda))$ |
| Our Algorithm$^\star$ (on well-spread $\mathcal{T}$) | $O(m)$ | $O(m)$ | $O(p \log_2 \lambda)$ | $O(p \log_2 \lambda)$ |

$^\star$ These algorithms recover Euclidean distances and others recover geodesic distances.

Table 1: Here "Time" is the time needed to embed a data point, while "Space" is the space needed to store the embedding matrix. "Storage" contains the memory usage to store each encoded sequence. "Query time" is the time complexity of pairwise distance estimation.

**Comparisons with baselines.** In Table 1, we compare our algorithm with various JL-based methods from Section 1. Here $n$ is the input dimension, $m$ is the embedding dimension (and number of bits), and $p = m/\lambda$ is the length of encoded sequences $\boldsymbol{y} = \widetilde{V} \boldsymbol{q}$. In our case, we use $O(p \log_2 \lambda)$ to store $\boldsymbol{y} = \widetilde{V} \boldsymbol{q}$. See Appendix E for a comparison with product quantization.

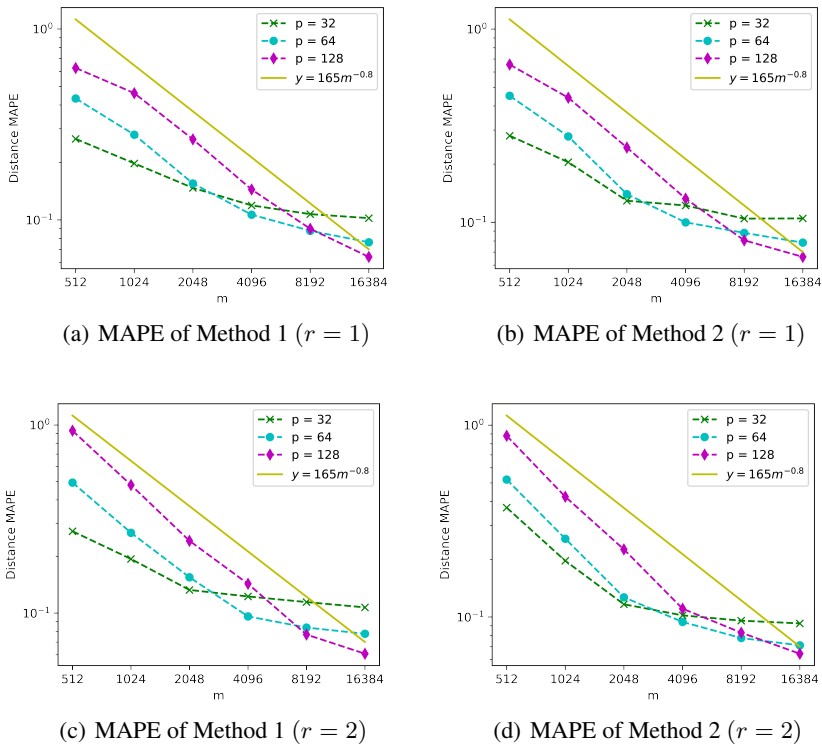

(a) MAPE of Method 1 ($r = 1$)  (b) MAPE of Method 2 ($r = 1$)

(c) MAPE of Method 1 ($r = 2$)  (d) MAPE of Method 2 ($r = 2$)

Figure 1: Plots of $\ell_2$ distance reconstruction error when $r = 1, 2$

## 6  NUMERICAL EXPERIMENTS

To illustrate the performance of our fast binary embedding (Algorithm 1) and $\ell_2$ distance recovery (Algorithm 2), we apply them to real-world datasets: Yelp open dataset[1], ImageNet (Deng et al., 2009), Flickr30k (Plummer et al., 2017), and CIFAR-10 (Krizhevsky et al., 2010). All images are converted to grayscale and resampled using bicubic interpolation to size $128 \times 128$ for images from Yelp, ImageNet, and Flickr30k and $32 \times 32$ for images from CIFAR-10. So, each can be represented by a $16384$-dimensional or $1024$-dimensional vector. The results are reported here and in Appendix A. We consider the two versions of our fast binary embedding algorithm from Theorem 4.2:

**Method 1.** We quantize FJLT embeddings $\mathbf{\Phi x}$, and recover distances based on Algorithm 2.

**Method 2.** We quantize sparse JL embeddings $\mathbf{A x}$ and recover distances by Algorithm 2.

In order to test the performance of our algorithm, we compute the mean absolute percentage error (MAPE) of reconstructed $\ell_2$ distances averaged over all pairwise data points, that is,

$$\frac{2}{k(k-1)} \sum_{\boldsymbol{x}, \boldsymbol{y} \in \mathcal{T}} \left| \frac{\|\widetilde{\boldsymbol{V}}(\boldsymbol{q_x} - \boldsymbol{q_y})\|_1 - \|\boldsymbol{x} - \boldsymbol{y}\|_2}{\|\boldsymbol{x} - \boldsymbol{y}\|_2} \right|.$$

**Experiments on the Yelp dataset.** To give a numerical illustration of the relation among the length $m$ of the binary sequences, embedding dimension $p$, and order $r$, as compared to the upper bound in (15), we use both Method 1 and Method 2 on the Yelp dataset. We randomly sample $k = 1000$ images and scale them by the same constant so all data points are contained in the $\ell_2$ unit ball. The scaled dataset is denoted by $\mathcal{T}$. Based on Theorem 4.2, we set $n = 16384$ and $s = 1650/n \approx 0.1$. For each fixed $p$, we apply Algorithm 1 and Algorithm 2 for various $m$. We present our experimental results for stable $\Sigma\Delta$ quantization schemes, given by (21), with $r = 1$ and $r = 2$ in Figure 1. For $r = 1$, we observe that the curve with small $p$ quickly reaches an error floor while with high $p$ the error decays like $m^{-1/2}$ and eventually reach a lower floor. The reason is that the first error term in (15) is dominant when $m/p$ is relatively small but the second error term eventually dominates as

---

[1]Yelp open dataset: https://www.yelp.com/dataset

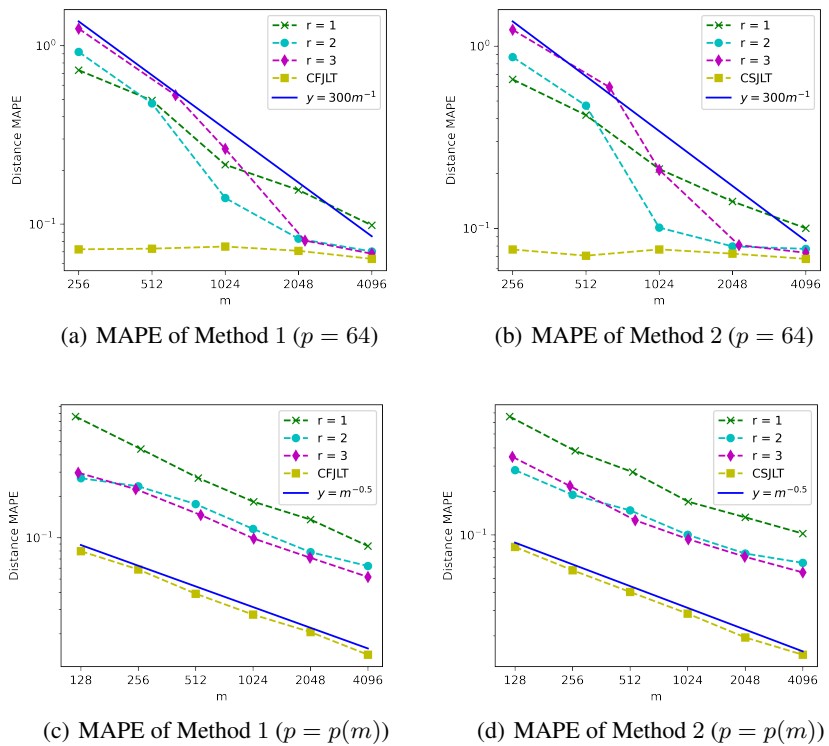

(a) MAPE of Method 1 ($p = 64$)  (b) MAPE of Method 2 ($p = 64$)

(c) MAPE of Method 1 ($p = p(m)$)  (d) MAPE of Method 2 ($p = p(m)$)

Figure 2: Plots of $\ell_2$ distance reconstruction error with fixed $p = 64$ and optimal $p = p(m)$

$m$ becomes larger and larger. When $r = 2$ the error curves decay faster and eventually achieve the same flat error because now the first term in (15) has power $-3/2$ while the second flat error term is independent of $r$. Moreover, the performance of Method 2 is very similar to that of Method 1.

Next, we illustrate the relationship between the quantization order $r$ and the number of measurements $m$ in Figure 2. The curves obtained directly from an unquantized CFJLT (resp. CSJLT) as in Lemma 2.6, with $m = 256, 512, 1024, 2048, 4096$, and $p = 64$ are used for comparison against the quantization methods. The first row of Figure 2 depicts the mean squared relative error when $p = 64$ is fixed for all distinct methods. It shows that stable quantization schemes with order $r > 1$ outperform the first order greedy quantization method, particularly when $m$ is large. Moreover, both the $r = 2$ and $r = 3$ curves converge to the CFJLT/CSJLT result as $m$ goes to 4096. Note that by using a quarter of the original dimension, i.e. $m = 4096$, our construction achieves less than 10% error. Furthermore, if we encode $\widetilde{V}q$ as discussed in Section 5, then we need at most $rp \log_2 \lambda = 64r \log_2(4096/64) = 384r$ bits per image, which is $\lesssim 0.023$ bits per pixel.

For our final experiment, we illustrate that the performance of the proposed approach can be further improved. Note that the choice of $p$ only affects the distance computation in Algorithm 2 and does not appear in the embedding algorithm. In other words, one can vary $p$ in Algorithm 2 to improve performance. This can be done either analytically by viewing the right hand side of (15) as a function of $p$ and optimizing for $p$ (up to constants). It can also be done empirically, as we do here. Following this intuition, if we vary $p$ as a function of $m$, and use the empirically optimal $p := p(m)$ in the construction of $\widetilde{V}$, then we obtain the second row of Figure 2 where the choice $r = 3$ exhibits lower error than other quantization methods. Note that the decay rate, as a function of $m$, very closely resembles that of the unquantized JL embedding particularly for higher orders $r$ (as one can verify by optimizing the right hand side of (15)).

### ACKNOWLEDGMENTS

Our work was supported in part by NSF Grant DMS-2012546 and a UCSD senate research award. The authors would like to thank Sjoerd Dirksen for inspiring discussions and suggestions.

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

## A    COMPARISONS ON DIFFERENT DATASETS

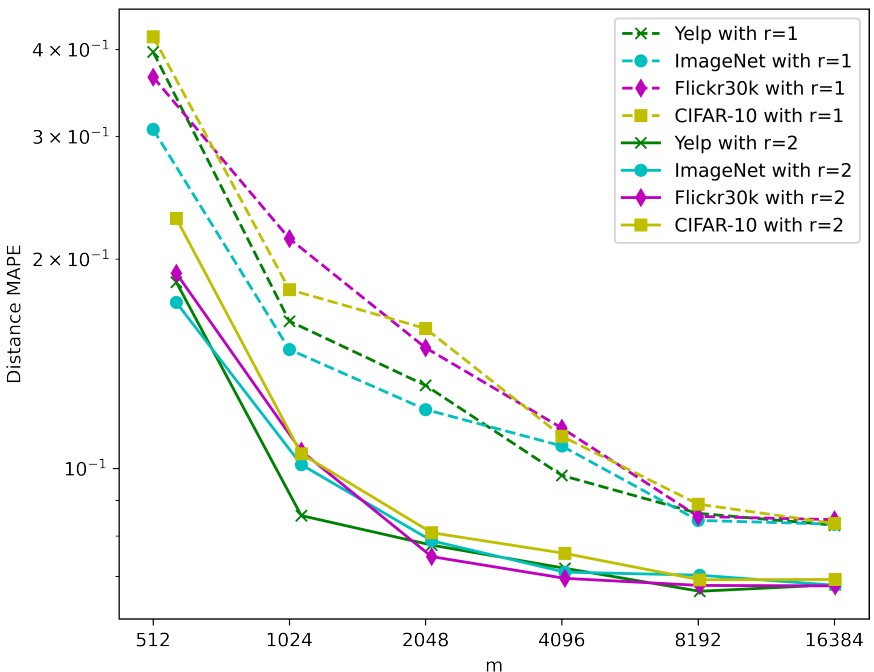

Figure 3: Plot of MAPE of Method 2 on four datasets with fixed $p = 64$ and order $r = 1, 2$

Experiments on the Yelp dataset in Section 6 showed that Method 2 based on sparse JL embeddings performs as well as Method 1 which uses an FJLT to enforce the well-spreadness assumption. Now, we only focus on Method 2 and check its performance on all four different datasets: Yelp, ImageNet, Flickr30k, and CIFAR-10.

Specifically, for each dataset we randomly sample $k = 1000$ images and scale them such that all scaled data points are contained in the $\ell_2$ unit ball. Then we apply Method 2 to each dataset separately and compute the corresponding MAPE metric, see Figure 3, where we fix $p = 64$ and let $r = 1, 2$. We can observe that curves with $r = 1$ fluctuate, but displays a clear downward trend, when $m \leq 8192$ and reach an error floor around $0.08$. In contrast to the first order quantization scheme, curves with $r = 2$ decays faster and eventually achieve a lower floor around $0.07$. Additionally, Method 2 performs well on all datasets and implies that assumption of well-spread input vectors is not too restrictive on natural images.

# B  PROOF OF LEMMA 2.6

We will require the following lemmas, adapted from the literature, to prove the distance-preserving properties of our condensed sparse Johnson-Lindenstrauss transform (CSJLT) and condensed fast Johnson-Lindenstrauss transform (CFJLT) in Lemma 2.6.

**Lemma B.1** (Theorem 5.1 in Matoušek (2008)). *Let $n \in \mathbb{N}$, $\epsilon \in (0, \frac{1}{2})$, $\delta \in (0, 1)$, $\alpha \in [\frac{1}{\sqrt{n}}, 1]$ be parameters and set $m = C\epsilon^{-2}\log(\delta^{-1}) \in \mathbb{N}$ where $C$ is a sufficiently large constant. Let $s = 2\alpha^2/\epsilon \leq 1$, $\mathbf{A} \in \mathbb{R}^{m \times n}$ be as in Definition 2.2. Then*

$$P\Big((1-\epsilon)\|\boldsymbol{x}\|_2 \leq \frac{\sqrt{\pi/2}}{m}\|\mathbf{A}\boldsymbol{x}\|_1 \leq (1+\epsilon)\|\boldsymbol{x}\|_2\Big) \geq 1 - \delta \tag{16}$$

*holds for all $\boldsymbol{x} \in \mathbb{R}^n$ with $\|\boldsymbol{x}\|_\infty \leq \alpha\|\boldsymbol{x}\|_2$.*

Lemma B.2 below is adapted from (Ailon & Chazelle, 2009, Lemma 1), and we present its proof for completeness.

**Lemma B.2.** *Let $\boldsymbol{H} \in \mathbb{R}^{n \times n}$ and $\mathbf{D} \in \mathbb{R}^{n \times n}$ be as in Definition 2.4. For any $\lambda > 0$ and $\boldsymbol{x} \in \mathbb{R}^n$ we have*

$$P\Big(\|\boldsymbol{H}\mathbf{D}\boldsymbol{x}\|_\infty \leq \lambda\|\boldsymbol{x}\|_2\Big) \geq 1 - 2ne^{-n\lambda^2/2}. \tag{17}$$

*Proof.* Without loss of generality, we can assume $\|\boldsymbol{x}\|_2 = 1$. Let $\boldsymbol{u} = \boldsymbol{H}\mathbf{D}\boldsymbol{x} = (u_1, \dots, u_n)$. Fix $i \in \{1, \dots, n\}$. Then $u_i = \sum_{j=1}^n a_j x_j$ with $P\Big(a_j = \frac{1}{\sqrt{n}}\Big) = P\Big(a_j = -\frac{1}{\sqrt{n}}\Big) = \frac{1}{2}$ for all $j$. Moreover, $a_1, a_2, \dots, a_n$ are independent and symmetric. So $u_i$ is also symmetric, that is, $u_i$ and $-u_i$ share the same distribution. For any $t \in \mathbb{R}$ we have

$$\mathbb{E}(e^{tnu_i}) = \prod_{j=1}^n \mathbb{E}[\exp(tna_j x_j)] = \prod_{j=1}^n \frac{\exp(t\sqrt{n}x_j) + \exp(-t\sqrt{n}x_j)}{2}$$

$$\leq \prod_{j=1}^n \exp(nt^2 x_j^2/2) = \exp(nt^2/2).$$

Since $u_i$ is symmetric, by Markov's inequality and the above result, we get

$$P(|u_i| \geq \lambda) = 2P(e^{\lambda n u_i} \geq e^{\lambda^2 n}) \leq 2e^{-\lambda^2 n}\mathbb{E}(e^{\lambda n u_i}) = 2e^{-\lambda^2 n/2}.$$

Inequality (17) follows by the union bound over all $i \in \{1, \dots, n\}$. $\square$

**Lemma B.3.** *Let $n, \lambda \in \mathbb{N}$, $\epsilon \in (0, \frac{1}{2})$, $\delta \in (0, 1)$, $p = O(\epsilon^{-2}\log(\delta^{-1})) \in \mathbb{N}$ and $m = \lambda p$. Let $\widetilde{\boldsymbol{V}} \in \mathbb{R}^{p \times m}$ be as in Definition 2.3, $\mathbf{A} \in \mathbb{R}^{m \times n}$ be the sparse Gaussian matrix in Definition 2.2 with $s = \Theta(\epsilon^{-1}n^{-1}(\|\boldsymbol{v}\|_\infty/\|\boldsymbol{v}\|_2)^2) \leq 1$, and $\boldsymbol{\Phi} = \mathbf{A}\boldsymbol{H}\mathbf{D} \in \mathbb{R}^{m \times n}$ be the FJLT in Definition 2.4 with $s = \Theta(\epsilon^{-1}n^{-1}(\|\boldsymbol{v}\|_\infty/\|\boldsymbol{v}\|_2)^2\log n) \leq 1$. Then for $\boldsymbol{x} \in \mathbb{R}^n$ with $\|\boldsymbol{x}\|_\infty = O(n^{-1/2}\|\boldsymbol{x}\|_2)$, we have*

$$\mathrm{P}\Big((1-\epsilon)\|\boldsymbol{x}\|_2 \leq \|\widetilde{\boldsymbol{V}}\mathbf{A}\boldsymbol{x}\|_1 \leq (1+\epsilon)\|\boldsymbol{x}\|_2\Big) \geq 1 - \delta, \tag{18}$$

*and for arbitrary $\boldsymbol{x} \in \mathbb{R}^n$, we have*

$$\mathrm{P}\Big((1-\epsilon)\|\boldsymbol{x}\|_2 \leq \|\widetilde{\boldsymbol{V}}\boldsymbol{\Phi}\boldsymbol{x}\|_1 \leq (1+\epsilon)\|\boldsymbol{x}\|_2\Big) \geq 1 - \delta. \tag{19}$$

*Proof.* Recall that $\boldsymbol{V} = \boldsymbol{I}_p \otimes \boldsymbol{v}$ and $\boldsymbol{\Phi} = \mathbf{A}\boldsymbol{H}\mathbf{D}$. Let $\boldsymbol{y} \in \mathbb{R}^n$ and $\mathbf{K} := \boldsymbol{V}\mathbf{A} = (\boldsymbol{I}_p \otimes \boldsymbol{v})\mathbf{A} \in \mathbb{R}^{p \times n}$. For $1 \leq i \leq p$ and $1 \leq j \leq n$, we have

$$K_{ij} = \sum_{k=1}^\lambda v_k a_{(i-1)\lambda+k, j}.$$

Denote the row vectors of $\mathbf{A}$ by $\mathbf{a}_1, \mathbf{a}_2, \dots, \mathbf{a}_m$. It follows that

$$(\mathbf{K}\boldsymbol{y})_i = \sum_{j=1}^n K_{ij}y_j = \sum_{j=1}^n \sum_{k=1}^\lambda y_j v_k a_{(i-1)\lambda+k, j} = \sum_{k=1}^\lambda v_k \langle \boldsymbol{y}, \mathbf{a}_{(i-1)\lambda+k}\rangle = [\mathbf{B}(\boldsymbol{v}^\mathsf{T} \otimes \boldsymbol{y})]_i$$

where

$$\mathbf{B} := \begin{bmatrix} \mathbf{a}_1 & \mathbf{a}_2 & \dots & \mathbf{a}_\lambda \\ \mathbf{a}_{\lambda+1} & \mathbf{a}_{\lambda+2} & \dots & \mathbf{a}_{2\lambda} \\ \vdots & \vdots & & \vdots \\ \mathbf{a}_{(p-1)\lambda+1} & \mathbf{a}_{(p-1)\lambda+2} & \dots & \mathbf{a}_{p\lambda} \end{bmatrix} \in \mathbb{R}^{p \times \lambda n} \quad \text{and} \quad \boldsymbol{v}^\top \otimes \boldsymbol{y} = \begin{bmatrix} v_1 \boldsymbol{y} \\ v_2 \boldsymbol{y} \\ \vdots \\ v_\lambda \boldsymbol{y} \end{bmatrix} \in \mathbb{R}^{\lambda n}.$$

Hence $\boldsymbol{V}\boldsymbol{A}\boldsymbol{y} = \boldsymbol{K}\boldsymbol{y} = \mathbf{B}(\boldsymbol{v}^\top \otimes \boldsymbol{y})$ holds for all $\boldsymbol{y} \in \mathbb{R}^n$. Additionally, we get a reshaped sparse Gaussian random matrix $\mathbf{B}$ by rearranging the rows of $\mathbf{A}$.

For the first assertion in the theorem, note that $\boldsymbol{x} \in \mathbb{R}^n$ satisfies $\|\boldsymbol{x}\|_\infty = O(\|\boldsymbol{x}\|_2/\sqrt{n})$. So, we have $\boldsymbol{V}\boldsymbol{A}\boldsymbol{x} = \mathbf{B}(\boldsymbol{v}^\top \otimes \boldsymbol{x})$, $\|\boldsymbol{v}^\top \otimes \boldsymbol{x}\|_2 = \|\boldsymbol{v}\|_2 \|\boldsymbol{x}\|_2$ and $\|\boldsymbol{v}^\top \otimes \boldsymbol{x}\|_\infty = \|\boldsymbol{v}\|_\infty \|\boldsymbol{x}\|_\infty$. Then (18) holds by applying Lemma B.1 to random matrix $\mathbf{B}$ and vector $\boldsymbol{v}^\top \otimes \boldsymbol{x}$ with $\alpha = \Theta(n^{-1/2}\|\boldsymbol{v}\|_\infty/\|\boldsymbol{v}\|_2)$.

For the second assertion, if $\boldsymbol{x} \in \mathbb{R}^n$ is arbitrary, then by substituting $\boldsymbol{H}\boldsymbol{D}\boldsymbol{x}$ for $\boldsymbol{y}$ one can get

$$\boldsymbol{V}\boldsymbol{\Phi}\boldsymbol{x} = \mathbf{B}(\boldsymbol{v}^\top \otimes (\boldsymbol{H}\boldsymbol{D}\boldsymbol{x})).$$

Note that

$$\|\boldsymbol{v}^\top \otimes (\boldsymbol{H}\boldsymbol{D}\boldsymbol{x})\|_2 = \|\boldsymbol{v}\|_2 \|\boldsymbol{H}\boldsymbol{D}\boldsymbol{x}\|_2 = \|\boldsymbol{v}\|_2 \|\boldsymbol{x}\|_2 \quad \text{and} \quad \|\boldsymbol{v}^\top \otimes (\boldsymbol{H}\boldsymbol{D}\boldsymbol{x})\|_\infty = \|\boldsymbol{v}\|_\infty \|\boldsymbol{H}\boldsymbol{D}\boldsymbol{x}\|_\infty.$$

Inequality (19) follows immediately by using the above fact and applying Lemma B.1 and Lemma B.2 to the random operator $\mathbf{B}$ and vector $\boldsymbol{v}^\top \otimes (\boldsymbol{H}\boldsymbol{D}\boldsymbol{x})$ with $\alpha = \Theta((n^{-1}\log n)^{1/2}\|\boldsymbol{v}\|_\infty/\|\boldsymbol{v}\|_2)$. $\square$

Now we can embed a set of points in a high dimensional space into a space of much lower dimension in such a way that distances between the points are nearly preserved. By substituting $\delta$ with $2\delta/|\mathcal{T}|^2$ in Lemma B.3 and using the fact $1 - \binom{|\mathcal{T}|}{2}\frac{2\delta}{|\mathcal{T}|^2} = 1 - \frac{|\mathcal{T}|(|\mathcal{T}|-1)}{2} \cdot \frac{2\delta}{|\mathcal{T}|^2} > 1 - \delta$, Lemma 2.6 follows from the union bound over all pairwise data points in $\mathcal{T}$.

## C   STABLE SIGMA-DELTA QUANTIZATION AND ITS PROPERTIES

Although it is a non-trivial task to design a stable quantization rule $\rho$ when $r > 1$, families of one-bit $\Sigma\Delta$ quantization schemes that achieve this goal have been designed by Daubechies & DeVore (2003); Güntürk (2003); Deift et al. (2011), and we now describe one such family. To start, note that an $r$-th order $\Sigma\Delta$ quantization scheme may also arise from a more general difference equation of the form

$$\mathrm{y} - \mathrm{q} = \boldsymbol{f} * \boldsymbol{v} \tag{20}$$

where $*$ denotes convolution and the sequence $\boldsymbol{f} = \boldsymbol{P}^r \boldsymbol{g}$ with $\boldsymbol{g} \in \ell^1$. Then any (bounded) solution $\boldsymbol{v}$ of (20) generates a (bounded) solution $\boldsymbol{u}$ of (11) via $\boldsymbol{u} = \boldsymbol{g} * \boldsymbol{v}$. Thus (11) can be rewritten in the form (20) by a change of variables. Defining $\boldsymbol{h} := \boldsymbol{\delta}^{(0)} - \boldsymbol{f}$, where $\boldsymbol{\delta}^{(0)}$ denotes the Kronecker delta sequence supported at 0, and choosing the quantization rule $\rho$ in terms of the new variable as $(\boldsymbol{h} * \boldsymbol{v})_i + y_i$. Then (10) reads as

$$\begin{cases} q_i = Q((\boldsymbol{h} * \boldsymbol{v})_i + y_i), \\ v_i = (\boldsymbol{h} * \boldsymbol{v})_i + y_i - q_i. \end{cases} \tag{21}$$

By designing a proper filter $\boldsymbol{h}$ one can get a stable $r$-th order $\Sigma\Delta$ quantizer, as was done in Deift et al. (2011); Güntürk (2003), leading to the following result from Güntürk (2003), which exploits the above relationship between $\boldsymbol{v}$ and $\boldsymbol{u}$ to bound $\|\boldsymbol{u}\|_\infty$.

**Proposition C.1.** *Fix an integer $r$, an integer $\sigma \geq 6$ and let $n_j = \sigma(j-1)^2 + 1$ for $j = 1, 2, \ldots, r$. Let the filter $h$ be of the form*

$$\boldsymbol{h} = \sum_{j=1}^r d_j \boldsymbol{\delta}^{n_j}$$

*where $\boldsymbol{\delta}^{n_j}$ is the Kronecker delta supported at $n_j$ and $d_j = \prod_{i \neq j} \frac{n_i}{n_i - n_j}$ for $j = 1, 2, \ldots, r$. There exists a universal constant $C > 0$ such that the $r$th order $\Sigma\Delta$ scheme (21) with 1-bit alphabet $\mathcal{A} = \{-1, 1\}$, is stable, and*

$$\|\boldsymbol{y}\|_\infty \leq \mu < 1 \Longrightarrow \|\boldsymbol{u}\|_\infty \leq Cc(\mu)^r r^r, \tag{22}$$

*where $c(\mu) > 0$ is a constant only depends on $\mu$.*

Having introduced stable $\Sigma\Delta$ quantization, we now present a lemma controlling an operator norm of $\widetilde{V}P^r$. We will need this result in controlling the error in approximating distances associated with our binary embedding.

**Lemma C.2.** *For a stable $r$-th order $\Sigma\Delta$ quantization scheme,*

$$\|\widetilde{V}P^r\|_{\infty,1} \leq \sqrt{\pi/2}(8r)^{r+1}\lambda^{-r+1/2}.$$

*Proof.* By the same method used in the proof of Lemma 4.6 in Huynh & Saab (2020), one can get

$$\|VP^r\|_{\infty,\infty} \leq r2^{3r-1} \quad \text{and} \quad \|v\|_2 \geq \lambda^{r-1/2}r^{-r}.$$

It follows that

$$\|\widetilde{V}P^r\|_{\infty,1} = \frac{\sqrt{\pi/2}}{p\|v\|_2}\|VP^r\|_{\infty,1} \leq \frac{\sqrt{\pi/2}}{\|v\|_2}\|VP^r\|_{\infty,\infty} \leq \sqrt{\pi/2}(8r)^{r+1}\lambda^{-r+1/2}.$$

$\square$

The following result guarantees that the linear part of our embedding generates a bounded vector, and therefore allows us to later appeal to the stability property of $\Sigma\Delta$ quantizers. In other words, it will allow us to use (22) to control the infinity norm of state vectors generated by $\Sigma\Delta$ quantization.

**Lemma C.3** (Concentration inequality for $\|\cdot\|_\infty$). *Let $\beta > 0$, $\epsilon \in (0,1)$, $\mathbf{A} \in \mathbb{R}^{m\times n}$ be the sparse Gaussian matrix in Definition 2.2 with $s = \Theta(\epsilon^{-1}n^{-1}) \leq 1$, and $\boldsymbol{\Phi} = \mathbf{A}H\mathbf{D} \in \mathbb{R}^{m\times n}$ be the FJLT in Definition 2.4 with $s = \Theta(\epsilon^{-1}n^{-1}\log n) \leq 1$. Suppose that*

$$2\sqrt{\beta + \log(2m)} \leq \mu \leq \frac{4}{\sqrt{\epsilon}}. \tag{23}$$

*Then*

$$P\big(\|\mathbf{A}x\|_\infty \leq \mu\|x\|_2\big) \geq 1 - e^{-\beta} \tag{24}$$

*holds for $x \in \mathbb{R}^n$ with $\|x\|_\infty = O(n^{-1/2}\|x\|_2)$ and*

$$P\big(\|\boldsymbol{\Phi}x\|_\infty \leq \mu\|x\|_2\big) \geq 1 - 2e^{-\beta} \tag{25}$$

*holds for $x \in \mathbb{R}^n$.*

*Proof.* Without loss of generality, we can assume that $x$ is a unit vector with $\|x\|_2 = 1$. We start with the proof of (25). By applying Lemma B.2 to $x$ with $\lambda = \Theta(\sqrt{\log n/n})$, we have

$$P\big(\|H\mathbf{D}x\|_\infty \leq \lambda\big) \geq 1 - e^{-\beta}. \tag{26}$$

Let $\mathbf{A}$ be as in Definition 2.2 with $s = 2\lambda^2/\epsilon = \Theta(\epsilon^{-1}n^{-1}\log n) \leq 1$ and recall that $\boldsymbol{\Phi} = \mathbf{A}H\mathbf{D}$. Suppose that $y \in \mathbb{R}^n$ with $\|y\|_2 = 1$ and $\|y\|_\infty \leq \lambda$. Let $\mathbf{Y} = \mathbf{A}y$. Then $Y_i := (\mathbf{A}y)_i = \sum_{j=1}^n a_{ij}y_j$ for $1 \leq i \leq m$. For $t \leq t_0 := \sqrt{2s}/\lambda = 2/\sqrt{\epsilon}$, we get $t^2 y_j^2/2s \leq 1$ for all $j$. Since $e^x \leq 1+2x$ for all $x \in [0,1]$ and $1+x \leq e^x$ for all $x \in \mathbb{R}$, $se^{t^2y_j^2/2s}+1-s \leq s(1+t^2y_j^2/s)+1-s = 1 + t^2y_j^2 \leq e^{t^2y_j^2}$. It follows that

$$\mathbb{E}(e^{tY_i}) = \prod_{j=1}^n \mathbb{E}(e^{ta_{ij}y_j}) = \prod_{j=1}^n \big(se^{t^2y_j^2/2s} + 1 - s\big) \leq \prod_{j=1}^n e^{t^2y_j^2} = e^{t^2}$$

holds for all $1 \leq i \leq m$ and $t \in [0,t_0]$. So for $t \in [0,t_0]$, by Markov inequality and above inequality we have

$$P\big(Y_i \geq \mu\big) = P\big(e^{tY_i} \geq e^{t\mu}\big) \leq e^{-t\mu}\mathbb{E}(e^{tY_i}) \leq e^{-t\mu+t^2}.$$

According to (23) we can set $t = \mu/2 \leq t_0 = 2/\sqrt{\epsilon}$, then $P\big(Y_i \geq \mu\big) \leq e^{-\mu^2/4}$. By symmetry we have $P\big(-Y_i \geq \mu\big) \leq e^{-\mu^2/4}$. Consequently, for all $1 \leq i \leq m$ we have

$$P\big(|Y_i| \geq \mu\big) \leq 2e^{-\mu^2/4}. \tag{27}$$

By a union bound, (23), and (27)

$$P(\|\mathbf{A}\boldsymbol{y}\|_\infty \geq \mu) = P\Big(\max_{1 \leq i \leq m} |Y_i| \geq \mu\Big) \leq mP\Big(|Y_i| \geq \mu\Big)$$

$$= 2me^{-\mu^2/4} \leq e^{-\beta}. \tag{28}$$

It follows immediately from (26) and (28) with $\boldsymbol{y} = \boldsymbol{H}\mathbf{D}\boldsymbol{x}$ that

$$
\begin{aligned}
P(\|\boldsymbol{\Phi}\boldsymbol{x}\|_\infty \leq \mu) &= P(\|\mathbf{A}\boldsymbol{H}\mathbf{D}\boldsymbol{x}\|_\infty \leq \mu) \\
&\geq P(\|\mathbf{A}\boldsymbol{H}\mathbf{D}\boldsymbol{x}\|_\infty \leq \mu, \|\boldsymbol{H}\mathbf{D}\boldsymbol{x}\|_\infty \leq \lambda) \\
&= P(\|\mathbf{A}\boldsymbol{H}\mathbf{D}\boldsymbol{x}\|_\infty \leq \mu \mid \|\boldsymbol{H}\mathbf{D}\boldsymbol{x}\|_\infty \leq \lambda)P(\|\boldsymbol{H}\mathbf{D}\boldsymbol{x}\|_\infty \leq \lambda) \\
&\geq (1 - e^{-\beta})^2 \\
&\geq 1 - 2e^{-\beta}.
\end{aligned}
$$

Furthermore, if we replace $\boldsymbol{y}$ by $\boldsymbol{x}$ in (28) and use $\mathbf{A}$ with $s = \Theta(\epsilon^{-1}n^{-1})$, then inequality (24) follows. The difference in the choice of $s$ is due to the fact that for vectors in the unit ball with $\|x\|_\infty = O(n^{-1/2}\|x\|_2)$ we have that $\|x\|_\infty \leq n^{-1/2}$. $\qquad\square$

## D    PROOF OF THEOREM 4.2

*Proof.* Since the proofs of (12) and (13) are almost identical except for using different random projections $\mathbf{A}$ and $\boldsymbol{\Phi}$, we shall only establish the result for (13) in detail. For any $\boldsymbol{x} \in \mathcal{T} \subseteq B_2^n(\kappa)$ we have $\|\boldsymbol{x}\|_2 \leq \kappa$. By applying Lemma C.3 we get

$$
\begin{aligned}
P(\|\boldsymbol{\Phi}\boldsymbol{x}\|_\infty < \mu) &\geq P(\|\boldsymbol{\Phi}\boldsymbol{x}\|_\infty < \mu\|\boldsymbol{x}\|_2/\kappa) \\
&\geq P(\|\boldsymbol{\Phi}\boldsymbol{x}\|_\infty < 2\sqrt{\beta + \log(2m)}\|\boldsymbol{x}\|_2) \\
&\geq 1 - 2e^{-\beta}.
\end{aligned}
$$

Since above inequality holds for arbitrary $\boldsymbol{x} \in \mathcal{T}$, by union bound one can get

$$P\Big(\max_{\boldsymbol{x} \in \mathcal{T}} \|\boldsymbol{\Phi}\boldsymbol{x}\|_\infty < \mu\Big) \geq 1 - 2|\mathcal{T}|e^{-\beta}.$$

Suppose that $\boldsymbol{u_x}$ is the state vector of input signal $\boldsymbol{\Phi}\boldsymbol{x}$ which is produced by stable $r$-th order $\Sigma\Delta$ scheme. Using Lemma C.2 and formula (22) to get

$$\|\widetilde{\boldsymbol{V}}\boldsymbol{P}^r\|_{\infty,1}\|\boldsymbol{u_x}\|_\infty \leq Cc(\mu)^r r^r (8r)^{r+1}\sqrt{\pi/2}\lambda^{-r+1/2}, \tag{29}$$

which holds uniformly for all $\boldsymbol{x} \in \mathcal{T}$ with probability exceeding $1 - 2|\mathcal{T}|e^{-\beta}$.

Furthermore, by Lemma 2.6 the probability that

$$\Big|\|\widetilde{\boldsymbol{V}}\boldsymbol{\Phi}(\boldsymbol{x} - \boldsymbol{y})\|_1 - \|\boldsymbol{x} - \boldsymbol{y}\|_2\Big| \leq \epsilon\|\boldsymbol{x} - \boldsymbol{y}\|_2 \tag{30}$$

holds simultaneously for all $\boldsymbol{x}, \boldsymbol{y} \in \mathcal{T}$ is at least $1 - \delta$.

We deduce from triangle inequality and equations (29), (30) that

$$
\begin{aligned}
&\Big|d_{\widetilde{\boldsymbol{V}}}(f_2(\boldsymbol{x}), f_2(\boldsymbol{y})) - \|\boldsymbol{x} - \boldsymbol{y}\|_2\Big| \\
&= \Big|\|\widetilde{\boldsymbol{V}}Q^{(r)}(\boldsymbol{\Phi}\boldsymbol{x}) - \widetilde{\boldsymbol{V}}Q^{(r)}(\boldsymbol{\Phi}\boldsymbol{y})\|_1 - \|\boldsymbol{x} - \boldsymbol{y}\|_2\Big| \\
&\leq \Big|\|\widetilde{\boldsymbol{V}}Q^{(r)}(\boldsymbol{\Phi}\boldsymbol{x}) - \widetilde{\boldsymbol{V}}Q^{(r)}(\boldsymbol{\Phi}\boldsymbol{y})\|_1 - \|\widetilde{\boldsymbol{V}}\boldsymbol{\Phi}(\boldsymbol{x} - \boldsymbol{y})\|_1\Big| + \Big|\|\widetilde{\boldsymbol{V}}\boldsymbol{\Phi}(\boldsymbol{x} - \boldsymbol{y})\|_1 - \|\boldsymbol{x} - \boldsymbol{y}\|_2\Big| \\
&\leq \|\widetilde{\boldsymbol{V}}(Q^{(r)}(\boldsymbol{\Phi}\boldsymbol{x}) - \boldsymbol{\Phi}\boldsymbol{x}) - \widetilde{\boldsymbol{V}}(Q^{(r)}(\boldsymbol{\Phi}\boldsymbol{y}) - \boldsymbol{\Phi}\boldsymbol{y})\|_1 + \Big|\|\widetilde{\boldsymbol{V}}\boldsymbol{\Phi}(\boldsymbol{x} - \boldsymbol{y})\|_1 - \|\boldsymbol{x} - \boldsymbol{y}\|_2\Big| \\
&\leq \|\widetilde{\boldsymbol{V}}\boldsymbol{P}^r\boldsymbol{u_x}\|_1 + \|\widetilde{\boldsymbol{V}}\boldsymbol{P}^r\boldsymbol{u_y}\|_1 + \Big|\|\widetilde{\boldsymbol{V}}\boldsymbol{\Phi}(\boldsymbol{x} - \boldsymbol{y})\|_1 - \|\boldsymbol{x} - \boldsymbol{y}\|_2\Big| \\
&\leq \|\widetilde{\boldsymbol{V}}\boldsymbol{P}^r\|_{\infty,1}(\|\boldsymbol{u_x}\|_\infty + \|\boldsymbol{u_y}\|_\infty) + \Big|\|\widetilde{\boldsymbol{V}}\boldsymbol{\Phi}(\boldsymbol{x} - \boldsymbol{y})\|_1 - \|\boldsymbol{x} - \boldsymbol{y}\|_2\Big| \\
&\leq 2Cc(\mu)^r r^r (8r)^{r+1}\sqrt{\pi/2}\lambda^{-r+1/2} + \epsilon\|\boldsymbol{x} - \boldsymbol{y}\|_2 \\
&= \sqrt{2\pi}Cc(\mu)^r r^r (8r)^{r+1}\lambda^{-r+1/2} + \epsilon\|\boldsymbol{x} - \boldsymbol{y}\|_2 \\
&= C(\mu, r)\lambda^{-r+1/2} + \epsilon\|\boldsymbol{x} - \boldsymbol{y}\|_2
\end{aligned}
$$

holds uniformly for any $x, y \in \mathcal{T}$ with probability at least $1 - \delta - 2|\mathcal{T}|e^{-\beta}$. The bound (12) is associated with a weaker condition on $\beta$ due to the associated weaker condition in Lemma C.3. $\square$

# E    COMPARISON WITH PRODUCT QUANTIZATION

Note that the distance preserving quality (as well as performance on retrieval and classification tasks) of MSQ binary embeddings using bilinear projection (Gong et al., 2013) or circulant matrices (Yu et al., 2014) has be shown to be at least as good as product quantization (Jegou et al., 2010), LSH (Andoni & Indyk, 2006; Shrivastava & Li, 2014) and ITQ (Gong et al., 2012c). Our method uses Sigma-Delta quantization, which

1. gives provably better error rates than the MSQ design as shown in this paper, and in (Huynh & Saab, 2020);

2. is more efficient in terms of both memory and distance query computation as shown in Section 5.

In order to more explicitly compare our algorithm with data dependent methods, as an example, we now briefly analyze product quantization as presented in (Jegou et al., 2010). We then present a brief analysis of optimal data-independent methods as well as data-independent product quantization, in comparison with our method.

DATA-DEPENDENT PRODUCT QUANTIZATION

The key idea here is to decompose the input vector space $\mathbb{R}^n$ into the Cartesian product of $M$ low-dimensional subspaces $\mathbb{R}^d$ with $n = Md$ and quantize each subspace into $k^*$ codewords, for example by using the $k$-means algorithm. So the total number of centroids (codewords) in $\mathbb{R}^n$ is $k = (k^*)^M$ and the time complexity of learning all $k$ centroids is $O(nNk^*t)$ where $N$ is the number of training data points and $t$ is the number of iterations in the $k$-means algorithm. Moreover, converting each input vector $x \in \mathbb{R}^n$ to the index of its codeword needs time $O(Mdk^*) = O(nk^*)$ and the length of binary codes is $m = \log_2 k = M \log_2 k^*$. Since we have to store all $k$ centroids and $M$ lookup tables, memory usage is $O(M(dk^* + (k^*)^2)) = O(nk^* + M(k^*)^2)$. Moreover, the query time, i.e. the time complexity of pairwise distance estimation is $O(Mk^*)$ using lookup tables. As a result, we obtain Table 2, whose column headings are analogous to those in Table 1.

| Method | Time | Space | Storage | Query Time |
|---|---|---|---|---|
| Product Quantization | $O(nk^*)$ | $O(nk^* + M(k^*)^2)$ | $O(M \log_2 k^*)$ | $O(Mk^*)$ |
| Our Method (on well-spread $\mathcal{T}$) | $O(m)$ | $O(m)$ | $O(p \log_2 \lambda)$ | $O(p \log_2 \lambda)$ |

Table 2: Comparison between the proposed method and product quantization per data point

A direct comparison of the associated errors is not possible due to the fact that the error associated with data-dependent product quantization is a function of the input data distribution, and the convergence of the $k$-means algorithm. Nevertheless, one can note some tradeoffs from Table 2. Namely, the embedding time and the space needed to store our embedding matrix are lower than those associated with product quantization. On the other hand, the space needed to store the embedded data points and the query time associated with product quantization depend on the parameter choices $M$ and $k^*$, which also affect the resulting accuracy. Finally, we note that product quantization (using $k$-means clustering) is associated with a pre-processing time $O(nNk^*t)$, which is significantly larger than our method.

DATA-INDEPENDENT PRODUCT QUANTIZATION AND OPTIMALITY OF OUR METHOD

If one were to just encode, in a data independent way, the $\ell_2$ ball of $\mathbb{R}^n$, so that the encoding error is at most $\theta$, then a simple volume argument shows that one needs at least $\theta^{-n}$ codewords, hence $n \log_2(1/\theta)$ bits. This lower bound holds, independent of the encoding method, i.e., whether one uses product quantization or any other technique. To reduce the number of bits below $n$, one

approach is to capitalize on the finiteness of the data, and use a JL type embedding (such as random sampling for well-spread data) to reduce the dimension to $p \approx \log |T|/\epsilon^2$ (up to log factors), and therefore introduce a new embedding error of $\epsilon$, on top of the encoding error. The advantage is that one would then only need to encode an $\ell_2$ ball in the $p$-dimensional space. Again, independently of the encoding method, one would now need $p \log(1/\theta)$ bits to get an encoding error of $\theta$. If we denote $c_x, c_y$, the encoding of $x$ and $y$, then this gives the error estimate

$$\left| \|c_x - c_y\| - \|x - y\| \right| \lesssim \theta + \epsilon \|x - y\|.$$

If we rewrite the error now in terms of the number of bits $b = p \log(1/\theta)$, we get

$$\left| \|c_x - c_y\| - \|x - y\| \right| \lesssim 2^{-b/p} + \epsilon \|x - y\|.$$

Note that in all of this, no computational complexity was taken into account.

One can envision replacing $k$-means clustering in product quantization, with a data-independent encoding. With a careful choice of parameters, this may be significantly more computationally efficient than the above optimal encoding, albeit at the expense of a sub-optimal error bound.

On the other hand, consider that our computationally efficient scheme uses $m$ bits, and that those $m$ bits can be compressed into $b \approx rp \log(m/p)$ bits (see Section 5), then our error, by Theorem 4.2 is

$$\left| \|c_x - c_y\| - \|x - y\| \right| \lesssim c(m/p)^{-r+1/2} + \epsilon \|x - y\|,$$

which in rate-distortion terms is

$$\left| \|c_x - c_y\| - \|x - y\| \right| \lesssim 2^{-\frac{b}{p}\frac{r-1/2}{r}} + \epsilon \|x - y\|.$$

In other words, up to constants in the exponent, and possible logarithmic terms, our result is near-optimal.

