# OpenReview forum: "Faster Binary Embeddings for Preserving Euclidean Distances"
_ICLR.cc/2021/Conference — ICLR 2021 Poster_

### Official Review · AnonReviewer1 · 2020-10-28
**Seems to be a straightforward variant of the prior work, perhaps I'm missing something**

**Rating:** 6
**Confidence:** 3

**Review:**

The paper studies binary embeddings that preserve Euclidean distances for the case when the vector mass is spread fairly evenly across the coordinates, which is a very common case in practice.

What the paper essentially does is a standard observation that uniform subsampling of coordinates (in spirit of Ailon -- Chazelle) of such dense vectors gives a Johnson--Lindenstrauss guarantee on the pairwise distances, and then it uses the quantization procedure developed earlier by Huynh and Saab.

To me the result sounds like a fairly straighforward ramification of the result of Huynh and Saab, but I can see it potentially being accepted, since the studied problem is extremely important.

I'm happy to revise the score if I got something wrong and the main resul is _not_ a straightforward variant of Huynh--Saab.

---

> ### Author Response · Authors · 2020-11-18
> **Detailed response**
>
> We thank the reviewer for the  feedback. We certainly agree that the studied problem is important! Below we address your comments in detail, and hope to convince you that there are sufficient contributions in this paper beyond the work of Huynh-Saab to merit acceptance. Here are some key differences with the work of Huynh and Saab:
>
> First, the previous result uses quantization method $q(x)=Q(DBx)$ where $q(x)$ is $m$-dimensional, with $O(n\log n)$ computational complexity and recovers original $l_2$ distances based on the $l_2$ norm of the $p$ dimensional vectors $y=Vq$. Thus distance computations need $O(p\log_2^2\lambda)$ time (where $\lambda=m/p$). In our paper, if we assume input vectors that are well-spread, we can reduce the time complexity of quantization to $O(m)$ by using $q(x)=Q(Gx)$ with sparse Gaussian matrix $G$. Additionally, we query the original Euclidean distance by using the $l_1$ norm of encoded sequences instead of $l_2$ norm, which further reduces the query time complexity from $O(p\log_2^2\lambda)$ to $O(p\log_2\lambda)$. The discussion related to time and space complexity can be found in Section 5 and we will add comparisons between our method and prior work in the revised version.
>
> Second, the proof of theoretical guarantees is rather different from previous work. The idea of replacing $DB$ by sparse Gaussian matrix $A$ (see formulas 3 and 5 in the paper) looks very natural and straightforward, but the proofs are distinct. In the work by Huynh and Saab, the proof is mainly based on the restricted isometry property (RIP) used for $l_2$ norms. Our proposed method uses the $l_1$ norm in the embedding space to recover the original $l_2$ norm, which implies that the RIP framework does not work for us.
>
> Third, unlike Huynh-Saab, we present numerical experiments in Section 6 to illustrate the relation of approximation error against the order of quantization $r$, embedding dimension $p$ and the length of binary code $m$. Numerical experiments are not presented in the paper by Huynh and Saab. We also note that based on other reviewer comments, extra numerical experiments have now been added, with  more datasets, including CIFAR-10, Flickr30K, and ImageNet to show that the method performs well for general natural images datasets.

---

### Official Review · AnonReviewer4 · 2020-10-29
**A new Algorithm with good theoretical guarantee, but the experiment is somewhat weak**

**Rating:** 7
**Confidence:** 2

**Review:**

In this paper, the author proposed a fast, distant-preserving, binary embedding algorithm. This is in contrast with most binary embedding methods in mapping and recovery. Their method is fast and memory efficient, and accurate, so that the error is compared to that of a continuous valued Johnson-Lindenstrauss embedding plus a quantization error. In addition to theoretical results, the authors also test the proposed algorithm on nature images, showing strong performance of their method.

The strong points includes
+Proposing a new binary embedding algorithm
+Providing theoretical results of approximate error
+Empirical results supporting the superiority of the proposed method

The weak points includes
-Some parts not easy to follow
-Experiments could be remarkably improved

Overall, the technical part looks strong, but the presentation and experiment could be improved.

First, the introduction part is difficult to follow, since too many symbols are used in this paper while some of which are not explained at all. Besides, in Lemma 2.6, it is unclear where to use $s$.

Second, the proposed algorithm is tested on one real-world dataset based on distance reconstruction error, and compared with FJLT. It could be much convincing if more datasets, more metrics, and more baselines are considered.

---

> ### Author Response · Authors · 2020-11-18
> **Detailed response**
>
> We would like to thank the reviewer for the  feedback. Below we address your comments and questions in detail.
>
> >First, the introduction part is difficult to follow, since too many symbols are used in this paper while some of which are not explained at all. Besides, in Lemma 2.6, it is unclear where to use $s$.
>
> Thanks for pointing out these issues. In the revised version, we give examples of sigma-delta quantization in the introduction, avoid forward looking references, and introduce other edits to improve readability.
>
> Regarding the variable $s$ in Lemma 2.6: $s$ is used to define the sparse Gaussian matrix $A$ in Definition 2.2. Then Definition 2.4 related to FJLT implicitly uses matrix $A$ to define $\Phi$ and thus uses variable $s$. In the Definition 2.4 of revised version, we will explicitly indicate the relation between $\Phi$ and $s$.
>
> >Second, the proposed algorithm is tested on one real-world dataset based on distance reconstruction error, and compared with FJLT. It could be much convincing if more datasets, more metrics, and more baselines are considered.
>
> The MAPE metric is used to evaluate/illustrate the theoretical upper bound indicated by Theorem 1.1 and Theorem 4.2.  The numerical experiments in Section 6, based on MAPE, illustrate the relation of approximation error against the order of quantization,  embedding dimension $p$ and the length of binary code $m$. We do take your point however, so we performed extra  numerical  experiments  on  more  datasets,  including  CIFAR-10, Flickr30K, and ImageNet.

---

### Official Review · AnonReviewer3 · 2020-10-29
**Another binary embedding of Euclidean space, but inadequate comparison with existing methods**

**Rating:** 6
**Confidence:** 5

**Review:**

The paper presents a new mapping of Euclidean vectors to bit vectors (quantization), along with a de-quantization method. The method is closely related to recent work by Huynh & Saab (2020), but instead of using a random rotation DBx on the input vector x, a sparse, Gaussian random projection is used. Under the assumption that the mass of x is "well spread out" this mapping also preserves Euclidean distances, but is quicker to compute (assuming random access to entries of x). After this mapping, a so-called Σ∆ quantization method is used, following Huynh & Saab.

The main issue I have with the paper is that there is no adequate comparison with quantization methods other than Σ∆ quantization. Notably, there is no comparison with product quantization (IEEE transactions on pattern analysis and machine intelligence, 2011). (Though it is presented as a data dependent method, it also makes sense as a data-independent method.) Also, LSH-based methods like simhash, and binarized E2LSH are not considered. It would also be helpful for the reader with a comparison to *data dependent* schemes, e.g. "Practical Data-Dependent Metric Compression with Provable Guarantees", NeurIPS 2017. Finally, there are related works in the theory literature that should probably be cited, e.g. "Optimal Compression of Approximate Inner Products and Dimension Reduction", FOCS 2017 (relevant for the case of normalized vectors).

Another claimed contribution, a speedup from O(n log n) to O(m) time, is not surprising (or new): It is known that for well spread vectors, even random sampling is an optimal dimension reduction method. (I think this goes back at least to Ailon and Chazelle.)

The writing can be made clearer. Even though I worked on related things, I found several parts of the paper hard to make sene of. For example:

- The abstract suggests that the results apply to all vectors that are not sparse, but in fact they apply only to vectors that satisfy a hard-to-satisfy L-infinity norm restriction.
- The abstract does not make clear that random access to the input vector is assumed, making it hard to understand how time less that O(n+m) is possible.
- Algorithms 1 and 2 in the introduction are impossible to understand without reading section 3 first. Maybe give a special case, with explicit details, and the general case later or in appendix?
- I suspect some assumption is missing in Theorem 1.1: The set of possible distances d_V(q_x,q_y) is finite, yet is supposed to be able to express the distance between arbitrary vectors x and y in R^n ...!
- The start of section 5 seems to imply that one can assume that vectors are well spread without loss of generality, by applying a random rotation. However, for random rotations the l-infinity norm will exceed the required norm by a factor sqrt(log n) with high probability.
- Is it important that L1 distance is used in algorithm 2, rather than L2 distance? After all, these distances are similar up to scaling for "spread out" vectors.
- Out of curiosity: Did you consider Kashin's representations of vectors?

---

> ### Author Response · Authors · 2020-11-18
> **Detailed response**
>
> We thank the reviewer for their thorough review, critical comments and valuable feedback. We have taken them all into account in editing our paper. Below are our responses to each of your comments.
>
> >The main issue I have with the paper is that there is no adequate comparison with quantization methods other than Sigma-Delta quantization. Notably, there is no comparison with product quantization (IEEE transactions on pattern analysis and machine intelligence, 2011). (Though it is presented as a data dependent method, it also makes sense as a data-independent method.) Also, LSH-based methods like simhash, and binarized E2LSH are not considered. It would also be helpful for the reader with a comparison to data dependent schemes, e.g. "Practical Data-Dependent Metric Compression with Provable Guarantees", NeurIPS 2017. Finally, there are related works in the theory literature that should probably be cited, e.g. "Optimal Compression of Approximate Inner Products and Dimension Reduction", FOCS 2017 (relevant for the case of normalized vectors).
>
> Thank you for raising this. We agree, and we now have a brief discussion on data dependent methods, including  product quantization, and LSH-based methods. That said, we feel that your review was rather harsh, and we note that
> 1. In the introduction section we cited  and mentioned several relevant papers in the *binary* quantization/embedding area, only one of which  (Huynh-Saab) uses $\Sigma\Delta$. Most of these prior works are built on a common framework (so we felt they were most relevant), that is, applying memoryless scalar quantization (MSQ) schemes or (in the case of Huynh-Saab) Sigma-Delta quantization to random projections of original data points and then recovering either Euclidean distances or geodesic distances based on encoded binary sequences.
> 2. In contrast, product quantization, and LSH-based methods may be associated with large time and space complexity for data preprocessing (i.e., performing the embedding). For example, to encode the data, product quantization needs to perform k-means clustering in each subspace to find potential centroids and also needs to store lookup tables. LSH-based methods need random shifts and (to our knowledge) dense random projections to quantize each input data point. Additionally, as you mentioned,  these methods are data dependent, which means that the accuracy of quantization depends on the underlying distribution of the input dataset. Of course one can imagine there are scenarios where one approach can be preferred over the other, but we think a much more detailed discussion would be outside the scope of our paper.
> 3. Having said that, useful comparisons do exist in the literature. For example, according to detailed comparisons in the paper ``On binary embedding using circulant matrices'' (JMLR, 2017), the distance preserving quality (as well as performance on downstream applications) of MSQ binary embeddings using circulant matrices can be as good or better than LSH and ITQ. We mention this because our method uses Sigma-Delta quantization, which gives provably better error rates than the MSQ design (as shown in this paper, and in Huynh-Saab, 2020).
>
> >Another claimed contribution, a speedup from O(n log n) to O(m) time, is not surprising (or new): It is known that for well spread vectors, even random sampling is an optimal dimension reduction method. (I think this goes back at least to Ailon and Chazelle.)
>
> We apologize for any confusion. We don't claim that we are inventing new sparse Johnson-Lindenstrauss embeddings. Recall that our goal is a fast *binary* embedding of the data -- we simply piggyback on fast JL methods, and inherit their speed guarantees. To prove our result, we do however show that certain matrices are JL matrices.
>
> Regarding your other point on random sampling, if the goal were to just embed the data into a lower dimensional space, then the reviewer would be  right that a random subsampling could work, provided the $\ell_2$ metric is used to compute distances between embedded points. On the other hand, using the $\ell_1$ metric when working with binarized data has some advantages for executing queries (lower bit complexity and lower computational complexity as a function of the number of bits used to represent the embedded vectors) -- please see Section 5. For these reasons, we rely on sparse Gaussians, which are marginally more computationally expensive than random subsampling. We have modified the introduction to clarify these points.

---

> > ### Author Response · Authors · 2020-11-18
> > **Detailed response part 2**
> >
> > >The abstract suggests that the results apply to all vectors that are not sparse, but in fact they apply only to vectors that satisfy a hard-to-satisfy L-infinity norm restriction.
> >
> > We do mention our assumption related to sparseness in the abstract, see the sentence ``When T consists of well-spread (i.e., non-sparse) vectors, our embedding method...'' Still, your point is well-taken, so we  edited the abstract to avoid ambiguity or confusion.
> >
> > Regarding the $\ell_\infty$ norm (well-spreadness) assumption, note that in practice, there are many settings where the well-spreadness assumptions is *not* too strong (natural images, random fourier features, subgaussian random variables). Similarly, there may be cases where the method works in practice, even though the theory requires the well-spreadness assumption. In the numerics section, we compare the performance of our method with and without using $HD$ to "flatten" the mass of the input vectors, see Figure 1 and Figure 2 in Section 6. The upshot is that there is barely any difference. Also, note that gaussian random vectors satisfy a well-spreadness assumption of the form $||x||_2 \leq C\frac{\log n}{\sqrt{n}}$ with high probability. In other words, most vectors drawn according to the gaussian measure are well-spread. One could easily modify our results to apply to such vectors, at the cost of a slightly increased (i.e., by a log factor) sparsity parameter in Theorem 4.2. We have now added a short discussion at the end of the introduction about the restrictiveness of the well-spreadness assumption.
> >
> > >The abstract does not make clear that random access to the input vector is assumed, making it hard to understand how time less that O(n+m) is possible.
> >
> > Perhaps the reviewer missed it, but we do have a sentence in the abstract: ``...our embedding method applies a stable noise-shaping quantization scheme to $Ax$ where $A\in\mathbb{R}^{m\times n}$ is a sparse Gaussian random matrix.'' Moreover, a discussion of the sparsity of $A$ can be found in Section 5.
> >
> > >Algorithms 1 and 2 in the introduction are impossible to understand without reading section 3 first. Maybe give a special case, with explicit details, and the general case later or in appendix?
> >
> > Thanks for pointing this out. We have now  provided an example of first order Sigma-Delta quantization in the introduction. Forward looking references will also be avoided as much as possible in the revised version.
> >
> > >I suspect some assumption is missing in Theorem 1.1: The set of possible distances $d_V(q_x,q_y)$ is finite, yet is supposed to be able to express the distance between arbitrary vectors x and y in $R^n$ ...!
> >
> > You are right that the set should be finite and contained in an $\ell_2$ ball. We now state this in the theorem. Please note that in  the  paper  we  assume  that  the  data  is   normalized, but only for ease of exposition.  If instead the data is such that the vectors are bounded in 2-norm by some constant, then all our results hold by using a binary alphabet of {-C,C}, instead of {-1,1}. The quantization part of the error bounds then simply scale by that constant.
> >
> > >The start of section 5 seems to imply that one can assume that vectors are well spread without loss of generality, by applying a random rotation. However, for random rotations the l-infinity norm will exceed the required norm by a factor sqrt(log n) with high probability.
> >
> > Thank you for catching this! We have modified the flattening lemmas, and corresponding results in Theorem 4.2, to account for the $\sqrt{\log n}$ term.
> >
> > Furthermore, considering that the current setting at the start of section 5 is a little bit confusing,  we will  remove or rephrase the sentence  ``...without loss of generality we may assume that T consists of well-spread vectors'' and just work with well-spread vectors in this section.
> >
> > >Is it important that L1 distance is used in algorithm 2, rather than L2 distance? After all, these distances are similar up to scaling for "spread out" vectors.
> >
> > This is a good question and the answer is yes. By using the $l_1$ norm of encoded sequences instead of $l_2$ norm, we reduce the query time complexity from $O(p\log^2\lambda)$ to $O(p\log\lambda)$. This is because computing an $\ell_1$ norm of a vector whose entries are $b$-bit integers entails only summing these integers, while computing an $\ell_2$ norm requires squaring these integers and computing a square root, which are more complex operations.  The discussion related to time and space complexity can be found in Section 5 where we have now added more details and comparisons between our method and prior work in the revised version.

---

> > > ### Author Response · Authors · 2020-11-18
> > > **Detailed response part 3**
> > >
> > > >Out of curiosity: Did you consider Kashin's representations of vectors?
> > >
> > > According to the paper ``Uncertainty principles and vector quantization'' by Yurii Lyubarskii and Roman Vershynin, a well-spread input vector $x$ can be expressed by Kashin’s coefficients using a tight frame in $\mathbb{R}^n$. Kashin’s representations can yield good robustness with respect to  uniform scalar quantization schemes, but this would mean one needs a frame, so $m>n$. It is also not clear whether it works well or has theoretical guarantees for euclidean distance computation (without first reconstructing the vectors), or whether it works well with noise-shaping methods such as Sigma-Delta quantization. We agree though that it may be interesting to pursue a connection in future work!

---

> > ### Comment · AnonReviewer3 · 2020-11-18
> > **Thoughtful revision**
> >
> > The authors have addressed most points that were not satisfactory, and the revised paper is in a much better state.
> >
> > I still think that a direct comparison with product quantization is needed to properly understand the contribution. While product quantization typically uses a data dependent quantization, it does not *need* to. In fact, if vectors are undergo a random rotation the quality of a data dependent quantization would probably be similar to a good, fixed quantization. It is also trivial to limit the output size of product quantization by simply stopping early. My recommendation is, therefore, to do a head-to-head comparison to product quantization using some good, fixed quantization methods, e.g. Golay codes.
> >
> > I am not sure that you understood my comment about random access. The point is that your algorithm depends on the vector x to be represented in a certain way (e.g. as an array). If it is stored, e.g., as a linked list, you cannot get running time below n.

---

> > > ### Author Response · Authors · 2020-11-20
> > > **Further Discussion**
> > >
> > > Thanks for the quick response. We are glad to see that our revised paper and comments address most of your concerns!
> > >
> > > >I still think that a direct comparison with product quantization is needed to properly understand the contribution. While product quantization typically uses a data dependent quantization, it does not need to. In fact, if vectors are undergo a random rotation the quality of a data dependent quantization would probably be similar to a good, fixed quantization. It is also trivial to limit the output size of product quantization by simply stopping early. My recommendation is, therefore, to do a head-to-head comparison to product quantization using some good, fixed quantization methods, e.g. Golay codes.
> > >
> > > This is an interesting point. The issue is the following: If one were to just encode, in a data independent way, the $\ell_2$ ball of $\mathbb{R}^n$, so that the encoding error is at most $\theta$, then one would need at least $\theta^{-n}$ codewords, hence $n\log(1/\theta)$ bits, so more bits than dimensions, which is too much for encoding a finite data set. This bound is true, independent of the encoding method, i.e., whether one uses product quantization or any other technique. One way out of this issue is to capitalize on the finiteness of the data, and use a JL type embedding (like random sampling for well-spread data) to reduce the dimension to $p \approx \log|T|/\epsilon^2$ (up to log factors), and therefore introduce a new
> > > embedding error of $\epsilon$, on top of the encoding error. The advantage is that one would then only need to encode the $p$-dimensional space. Again, independently of the encoding method, one would now need $p \log(1/\theta)$ bits to get an encoding error of $\theta$. If we denote $c_x,c_y$, the encoding of  $x$ and $y$, then this gives the error estimate
> > > $$\big| ||c_x-c_y||-||x-y|| \big| \lesssim \theta + \epsilon ||x-y||. $$ If we rewrite the error now in terms of the number of bits $b=p\log(1/\theta)$, we get
> > > $$\big| ||c_x-c_y||-||x-y|| \big| \lesssim 2^{-b/p} + \epsilon ||x-y||. $$
> > > Note that in all of this, no computational complexity was taken into account. This is just the best that one can do under any encoding (if a JL embedding is used).
> > > Consider that our scheme uses $m$ bits, and that those $m$ bits can be compressed into $b \approx r p\log(m/p)$ bits (see Section 5), then our error is $$\big| ||c_x-c_y||-||x-y|| \big| \lesssim c(m/p)^{-r+1/2} + \epsilon ||x-y||, $$ which in rate-distortion terms is
> > > $$\big| ||c_x-c_y||-||x-y|| \big| \lesssim 2^{-\frac{b}{p}\frac{r-1/2}{r}} + \epsilon ||x-y||. $$
> > > In other words, up to constants in the exponent, and possible log terms, our result is about as good as it can get.
> > >
> > > Using product quantization instead of the (computationally inefficient, and memory intensive) optimal encoding, would improve the computational complexity and memory tradeoffs (particularly if a Leech lattice or Golay code is used), at the expense of a sub-optimal error bound. We feel that it would be difficult and distracting to discuss this in the paper (since we would need to make all our above estimates precise), but we hope that this discussion is helpful in clarifying our contribution further!
> > >
> > > >I am not sure that you understood my comment about random access. The point is that your algorithm depends on the vector x to be represented in a certain way (e.g. as an array). If it is stored, e.g., as a linked list, you cannot get running time below n.
> > >
> > > Thanks for the clarification. We understand what you were saying now, and we definitely agree that it would not be possible to obtain O(m) time with something like a linked list. We do (implicitly) assume that either the data is being embedded into $m$ dimensions at the data acquisition stage (as in the compressed sensing model), or that it it stored in an array format.

---

> > > > ### Comment · AnonReviewer3 · 2020-11-20
> > > > **Further Discussion**
> > > >
> > > > It is true that there are nontrivial choices to make when deploying product quantization (whether on the original vectors or following a dimension reduction step), and I agree that this approach is likely to have better speed and higher error. It might be too much to discuss this in detail in the present manuscript, but I think a reader would like at least a hint of what this different trade-off looks like. Is it possible to have much better speed (for decoding), with only *slightly* higher error? A simple possibility to investigate would be a quantization to a random set of 2^b vectors for small-ish b.

---

> > > > > ### Author Response · Authors · 2020-11-25
> > > > > **Further Discussion**
> > > > >
> > > > > A new modified version of the paper has been posted where appendix E was added to compare our method with product quantization in both data-dependent and data-independent scenarios. Please note that we highlighted the major changes in blue for convenience.
> > > > >
> > > > > Regarding getting much better speed for decoding (compared with optimal encoding),  with only a slightly higher error using product quantization, this may be possible with careful selection of parameters of product quantization.  However, it remains true that having $2^b$ vectors for encoding a $p$-dimensional ball (regardless of whether they are random or not) automatically means a worst case error of $2^{-b/p}$ (again, using a volume argument). We allude to this, and note that our method approaches this bound, in Appendix E.

---

### Official Review · AnonReviewer2 · 2020-11-02
**A solid, but fairly incremental contribution**

**Rating:** 7
**Confidence:** 4

**Review:**

This paper proposes a scheme for fast distance-preserving binary embeddings of R^n into {-1/+1}^m. Guarantees are given for embedding l2-distance in the host space. Under certain assumptions about the input data both encoding and decoding can be done in O(m) time/space, which improves substantially upon the O(n log n) complexity from the previous work. Compared to the previous work by Huynh and Saab (2020) and Chou and Gunturk (2016) the key idea is to apply condensed JL transform instead of the Sigma-Delta quantization or other quantization methods used by the previous work. This is a very natural idea and it is somewhat surprising that this hasn’t been done before.

The overall technical result is highly technical to state (Theorem 4.2) so I won’t attempt to fully reproduce it here. On a technical level the results in this paper are hardly too surprising for the JL community, but it is nice to see this analysis worked out in detail. The key weakness of this paper is that in order to get optimal time/space complexity strong assumptions on the input vectors are required. The well-spread assumption used in the paper roughly corresponds to saying that the input vectors look like +/- O(1) in most coordinates, which might be unrealistic in the case of inputs with a small number of large coordinates. Despite this shortcoming, I think the paper is still going to be of moderate interest to the theoretical community.\

Overall, the paper is clearly written and presents and contributions well within the landscape of the previous work.

Other comments:

-- Maybe I missed it but it might be useful to add some discussion regarding the optimality of the first term in the bound in Theorem 1.1, i.e. is this term required?

-- Enlarge images and font sizes in Figure 2

-- Experiments are done one small data (500 128x128 images) -- this is not very convincing for a big data paper.

-- Is MAPE the right metric for image compression?

---

> ### Author Response · Authors · 2020-11-18
> **Detailed response**
>
> We thank the reviewer for the comments and interest in our paper. Below we address your comments and questions.
>
> >The key weakness of this paper is that in order to get optimal time/space complexity strong assumptions on the input vectors are required. The well-spread assumption used in the paper roughly corresponds to saying that the input vectors look like +/- O(1) in most coordinates, which might be unrealistic in the case of inputs with a small number of large coordinates.
>
> Thanks for giving us the opportunity to clarify this. Regarding the issue of well-spread vectors, we would like to make 3 points. First, we are aware that in some applications, one does not have such vectors, so we made it a point to highlight (the known fact) that one can transform vectors into well-spread vectors by applying something like a randomized Walsh-Hadamard transform. Even with this modification, which we are aware changes the computational complexity back to $O(n \log n)$ (as we state in the abstract and introduction), one important benefit of the proposed method is maintained: Euclidean distances can still be approximated by an $\ell_1$ norm, as opposed to an $\ell_2$ norm, on the binary embedded data. This implies the bit-complexity of the distance computation is reduced. We tried to make this point in Section 5 and we have added  language in the paper to clarify these points.
>         Our second point is that in practice there are many settings where the well-spreadness assumptions is *not* too strong, and our theoretical results would be useful in those settings. Based on other reviewers' requests, we have now added a short discussion of this at the end of the introduction. There may also be cases where the proposed method works well in practice, even though the theory requires the well-spreadness assumption. In the numerics section, we compare the performance of our method with and without using $HD$ to "flatten" the mass of the input vectors, see Figure 1 and Figure 2 in Section 6. The upshot is that there is barely any difference. As a third point, note that gaussian (and subgaussian) random vectors satisfy a well-spreadness assumption of the form $||x||_2 \leq C\sqrt{\frac{\log n}{n}} ||x||_\infty$ with high probability. In other words, most vectors drawn according to the gaussian measure are well-spread. One could easily modify our results to apply to such vectors, at the cost of a slightly increased (i.e., by a log factor) sparsity parameter in Theorem 4.2.
>
> >Maybe I missed it but it might be useful to add some discussion regarding the optimality of the first term in the bound in Theorem 1.1, i.e. is this term required?
>
> The first term $c(m/p)^{-r+1/2}$ in the bound is due to the quantization error and the second term results from the condensed JL embedding (Lemma 2.6). Since we are approximating the upper bound of the error caused by binary quantization in Theorem 1.1, the first term is required. It is a good open question whether the error bound can be replaced by a multiplicative error bound instead of the additive one we have. Regarding optimality, instead of polynomial decay in $m$, one would like to obtain an exponentially decaying bound. This can likely be done by using a distributed noise-shaping quantizer as in (Huynh, Saab 2020). Having said that, we also refer the reviewer to the final paragraph of the numerical experiments section, and to figure 2, which shows that by a careful choice of $p$ and $r$, one can obtain (up to constants and log factors) an error rate that closely matches that of an *unquantized* JL embedding.

---

> > ### Author Response · Authors · 2020-11-18
> > **Detailed response part 2**
> >
> > >(1) Enlarge images and font sizes in Figure 2. (2) Experiments are done one small data (500 128x128 images) -- this is not very convincing for a big data paper. (3) Is MAPE the right metric for image compression?
> >
> > (1) Thank you for this comment. We enlarged them as much as possible (subject to the page limit). (2) We checked that the  results are almost identical when we increase the number of images significantly, for several of our parameter choices. However, since we are running all our experiments on our local machines, we are somewhat limited in how large the datasets we can handle are (especially since we compute and compare all pairwise distances, including original ones and recovered ones). Nonetheless, your point is well-taken, and we will increase the size of our dataset as much as possible in the revised paper (we have already modified our experiments, doubling the size of our datasets). (3) The MAPE metric is chosen to be consistent with the theoretical results in Theorem 1.1 and Theorem 4.2. The numerical experiments in Section 6, based on MAPE, are intended to clearly illustrate the relation of approximation error against the order of quantization $r$, embedding dimension $p$ and the length of binary code $m$. Of course, if our algorithm is applied to k-nearest neighbors algorithm or image retrieval, one can compare the recall rate against different $m$ (length of binary codes), but we chose to separate the distance preserving binary embedding, from downstream applications so we can focus on the former in the paper.

---

### Official Review · AnonReviewer5 · 2020-11-07
**Fast Binary Embeddings**

**Rating:** 5
**Confidence:** 5

**Review:**

The authors present a distance preserving embedding algorithm to reduce the dimensionality / encoding of a high dimensional Euclidean point-set. The proposed embedding is a combination of stable noise-shaping quantization and sparse Johnson-Lindenstrauss transformations. The proposed method requires O(m) time and space complexity. The main contribution of the paper is Theorem 1.1 where the authors prove a bound on the distortion of the proposed embedding. The (additive) distortion consist of two terms due to the quantization error and JL relative error.


Reasons for score:
Overall, I vote for a slightly below acceptable due to the following concerns: (a) after reading the paper it was not clear if the authors do assume that the input pointset are well-spread or not. Theorem 1.1 states that the input points are well-spread and moreover Algorithm 1 assumes (implicitly) that the input is normalized. (b) the time complexity of the embedding is O(m) for well-spread vectors. In the first sentence of Section 5, the authors state that well-spread vectors can be assumed after an fast JL transformation. Indeed, but then the running time is not O(m), right? To transform any point-set to well-spread position you need at least O(nlogn).

Strong points:
* Well written paper with a clear contribution statement
* Concise algorithm description and corresponding theoretical guarantees.

Concerns:
* References to input sparsity time embedding (see below) are missing. How does your results compare to these papers?
* Theorem 1.1: Is the main contribution here the quantization part of the statement? It is known that the input pointset can be efficiently projected to p dimensions. Is it possible to decouple your contribution on quantization with JL projections?
* Theorem 1.1: Isn't the assumption of well-spreadness too strong here? If the vectors are so well-spread, I believe that uniformly sampling of coodinates could also work. Please discuss it in the paper.

Minor comments:
*Introduction: what is an $\epsilon$-Lipschitz distortion? Do you mean (1+\eps) distortion?
*Introduction: You may want to compare/discuss your work with input sparsity embeddings by Woodruff and Clarkson "Low Rank Approximation and Regression in Input Sparsity Time" and relevant count-min sketch embeddings.
* Algorithm 1: does algorithm 1 requires scaled data points or is this required only for the analysis?
* Equation (20) is referenced quite early. Please consider introducing it earlier.
* Theorem 1.1: "with high probability" -> this can be made more explicit using the \delta parameter as in the appendix.
* In general, there are several forward looking references in the text ("Finally, Definition 2.3 shows..", "Equation (20)". Please minimize such forward references.
* Theorem 4.2: Isn't it better to fix \beta to be greater than O(ln |T| / \delta). Otherwise the probability statements involve negative probabilities.
* Section 5, first sentence: rephrase the "without of loss of generality" statement.

---

> ### Author Response · Authors · 2020-11-18
> **Detailed response**
>
> We thank the reviewer for the insightful questions and great suggestions in terms of related work and potential improvement. Below, we provide a response that will hopefully address your comments, questions, and concerns:
>
> >After reading the paper it was not clear if the authors do assume that the input pointset are well-spread or not. Theorem 1.1 states that the input points are well-spread and moreover Algorithm1 assumes (implicitly) that the input is normalized.
>
> Regarding  normalization, in the paper we assume that the data is indeed normalized,  but only for ease of exposition. If the data is such that the vectors are bounded in 2-norm by some constant, then all our results hold by using a binary alphabet of {-C,C}, instead of  {-1,1}. The quantization component of the error bounds then simply scales by that constant.
>
> As for the issue of the vectors being well-spread, note that Theorem 4.2 considers both cases: arbitrary vectors, leading to $O(n\log n)$ embedding time, and well-spread vectors leading to $O(m)$ embedding time. Being aware that in some applications, one may not have well-spread vectors,  we made it a point to highlight (the known fact) that one can transform vectors into well-spread vectors by applying something like a randomized Walsh-Hadamard transform, and to show that our error bounds still hold. Even with this modification, which changes the computational complexity to $O(n \log n)$, one important benefit of the proposed method is maintained: Euclidean distances can still be approximated by an $\ell_1$ norm on the binary embedded data. This implies the bit-complexity of the distance computation is reduced (from $p\log^2(m/p)$ to $p\log(m/p)$ bits).  We made this point in Section 5. Moreover, we modified the abstract and introduction to clarify  that when the vectors are not well-spread the computational complexity for embedding a point is $O(n\log n)$. Please also see our response to the second point of part 2 below (which seemed like a distinct issue), regarding well-spread vectors.
> To summarize:
> 1. We assume that we have a high-dimensional finite dataset $\mathcal{T}\subset\mathbb{R}^n$ consisting of well-spread vectors in the unit ball. If it is not well-spread, we can transform it by applying a Walsh-Hadamard transform (at an increased preprocessing cost of $O(n\log n)$ per point). If it is not in the unit ball, we can either scale the data, or scale the quantization alphabet (which would be easier).
> 2. Then we apply a sparse Gaussian random projection $A$ to each  data point $x$ and use Sigma-Delta quantizers to quantize $Ax$, i.e., to obtain  $q_x=Q(Ax)$, which is Algorithm 1 in the paper. This concludes the binary embedding stage.
> 3. Finally, at query time, one can use Algorithm 2 to recover the original Euclidean distance from the binarized sketches $q_x$, via an $\ell_1$-norm calculation on integer vectors, with low computational and bit complexity.
>
> > In the first sentence of Section 5, the authors state that well-spread vectors can be assumed after an fast JL transformation. Indeed, but then the running time is not O(m), right?
>
> You are right. Hopefully our response above helps address your concern. Moreover, we have modified the first paragraph of Section 5 to assume that all input vectors are well-spread in this section.
>
> >References to input sparsity time embedding (see below) are missing. How does your results compare to these papers?
>
> We checked the results in "Low Rank Approximation and Regression in Input Sparsity Time" by Woodruff and Clarkson. Note that their algorithms are designed to improve least-squares regression, low-rank approximation and $l_p$-regression, and do not focus on *binary* embeddings, which is where our main contribution lies. Nevertheless, we will cite their work in our revised version. To be clear, we make no claims about inventing new sparse embeddings, please see our response below.

---

> > ### Author Response · Authors · 2020-11-18
> > **Detailed response part 2**
> >
> > >Theorem 1.1: Is the main contribution here the quantization part of the statement? It is known that the input pointset can be efficiently projected to p dimensions. Is it possible to decouple your contribution on quantization with JL projections?
> >
> > In a sense, yes (to both questions).  The main contribution of the paper is first in constructing a binary embedding of the data that can be computed fast and, second, in using the binary embedding to compute Euclidean distances with low computational complexity and bit complexity, and with provable accuracy guarantees. To that end, from a technical standpoint, we needed  to prove a new ``condensed'' JL lemma (see Lemma 2.6 in the paper) to show that original data points can be embedded into p-dimensional space. This then allowed us to combine the new condensed JL lemma and Sigma-Delta quantization together to generate the upper bound in Theorem 1.1, which is not based on the well-known classic JL lemma. From an algorithmic point of view, we make no novelty claims about the use of sparse JL transforms, just that we demonstrate they can be used in conjunction with noise-shaping quantizers to obtain fast and accurate binary embedding algorithms.
> >
> > > Isn't the assumption of well-spreadness too strong here? If the vectors are so well-spread, I believe that uniformly sampling of coodinates could also work. Please discuss it in the paper.
> >
> > Thank you, we will add a discussion in the paper. In addition to our above comments about well-spreadness, note that in practice, there are many settings where the well-spreadness assumptions is *not* too strong (e.g., generic feature vectors, natural images). Similarly, there may be cases where the method works in practice, even though the theory requires the well-spreadness assumption. For example, in the numerics section, we compare the performance of our method with and without using $HD$ to "flatten" the mass of the input vectors, see Figure 1 and Figure 2 in Section 6. The upshot is that there is barely any difference. Also, note that gaussian (and subgaussian) random vectors, as a stand in for generic vectors, satisfy a well-spreadness assumption of the form $||x||_2 \leq C\frac{\log n}{\sqrt{n}} ||x||_\infty$ with high probability. In other words, most vectors drawn according to the gaussian measure are well-spread. One could easily modify our results to apply to such vectors, at the cost of a slightly increased (i.e., by a log factor) sparsity parameter in Theorem 4.2.
> >
> > Regarding your question on random sampling of coordinates, recall that the goal is a *binary* embedding of the data. If the goal were to just embed the data into a lower dimensional space, then the reviewer would be  right that a random subsampling could work, provided the $\ell_2$ metric is used to compute distances between embedded points and provided the dimension of the embedding grows poly-logarithmically in the number of data points. On the other hand, as we discuss in Section 5, using the $\ell_1$ metric when working with binarized data has some advantages (lower bit complexity, lower computational complexity as a function of the number of bits used to represent the vectors). For this reason, we rely on sparse Gaussians, which are marginally more computationally expensive than random subsampling, but enable working with $\ell_1$ norms. Given space limitations it is difficult to discuss all these  points in detail in the paper, but we made a concise statement to this effect in the introduction.

---

> > > ### Author Response · Authors · 2020-11-18
> > > **Response to minor comments**
> > >
> > > >Q: What is an $\epsilon$-Lipschitz distortion? Do you mean $(1+\epsilon)$ distortion?
> > >
> > > R: Yes, indeed, an $\epsilon$-Lipschitz distortion means $(1+\epsilon)$ distortion.
> > >
> > > >Q: Does algorithm 1 requires scaled data points or is this required only for the analysis?
> > >
> > > R: Please see our response to your very first point above.
> > >
> > > >Concern: Equation (20) is referenced quite early. Please consider introducing it earlier.
> > >
> > > R: Thank you for pointing this out. We agree, we now present the first order $\Sigma\Delta$ quantization in the introduction section as an example, and refer to it in the algorithm.
> > >
> > > >Concern: Theorem 1.1: ``with high probability" -> this can be made more explicit using the $\delta$ parameter as in the appendix.
> > >
> > > R: Agreed, and done.
> > >
> > > > Concern: In general, there are several forward looking references in the text ("Finally, Definition 2.3 shows..", "Equation (20)". Please minimize such forward references.
> > >
> > > R: Thanks for pointing this out. We tried to do this, however we couldn't remove all instances as that would make the introductory section much too long, and push the main results too far back in the paper.
> > >
> > > >Concern: Theorem 4.2: Isn't it better to fix $\beta$ to be greater than $O(ln |T| / \delta)$. Otherwise the probability statements involve negative probabilities.
> > >
> > > R: Thank you for making this good point. We have made the change.
> > >
> > > >Concern: Section 5, first sentence: rephrase the "without of loss of generality" statement.
> > >
> > > R: Done. We now only assume well spread vectors in Section 5.

---

### Author Response · Authors · 2020-11-18
**The revised paper has been uploaded!**

We would like to thank all reviewers and AC for their hard work and valuable feedback. Your comments were very helpful and we took them all into account as we modified our manuscript to address the highlighted issues. We also provide a point-by-point response to all reviewers.

**A modified version of the paper has been posted to reflect the suggestions from the reviewers. Please note that we highlighted the major changes in blue for convenience.**

Summary of major changes in the manuscript:
1. Added extra numerical experiments on more datasets, including CIFAR-10, Flickr30k, and ImageNet to show that our method performs well for general natural images datasets. See Section 6 and Appendix A for details.
2. Corrected typos, modified Theorem 4.2, Lemma C.3, and rephrased Theorem 1.1 for clarity.
3. Added references and a brief discussion of data-dependent embedding methods, including product quantization, LSH-based methods and iterative quantization.
4. In Section 5, we compare our algorithm with various JL-based methods outlined in Section 1 from the perspective of time and space complexity.
5. Added a brief discussion of normalization issues, as well the well-spreadness assumption in section 1.2.

---

> ### Author Response · Authors · 2020-11-25
> **The revised paper (version 2) has been uploaded!**
>
> We would like to once more thank the Area Chair and Reviewers for their feedback. We have taken all your feedback into account in our revised version, and hope to have addressed all your points. We would especially like to thank Reviewer 3 for engaging us in a stimulating discussion. We believe our paper is better for it.

---

### Decision · Program_Chairs · 2021-01-07
**Final Decision**

**Decision:**

Accept (Poster)

**Comment:**

The paper provides a new distance preserving embedding based on a recent result called sigma-delta quantization. The authors notice that in many realistic scenarios, the input vectors are well-spread and under assumptions regarding the spreadness provide a fast technique to convert the input vectors into binary vectors, possibly of lower dimension. For completeness, the authors analyze the setting where the vectors are not spread and show that by using a randomized Walsh-Hadamard transform, their results still apply.
The authors do not provide a completely novel approach, to quote R2 “On a technical level the results in this paper are hardly too surprising for the JL community, but it is nice to see this analysis worked out in detail”. That being said, they show that a natural idea indeed works out by providing both a theoretical analysis and experimental results. The experiments can be more thorough but do convey the point that the result indeed works and moreover is somewhat robust in that it works well even when the formal requirements do not entirely hold.
There are a few issues mentioned by the reviewers that should be addressed: A clearer exposition of the guarantees and assumptions, some comparison with previous papers. However given the responses and discussions these seem minor and fixable towards a camera ready version. I recommend accepting the paper